# Extracting Weather Information from a Plantation Document

Gregory Burris[1], Jane Washburn[1], Omar Lasheen[1], Sophia Dorribo[1], James B. Elsner[1], and Ronald E. Doel[1]

[1]Florida State University, 113 Collegiate Loop, Tallahassee, FL 32306

**Correspondence:** Gregory Burris (gdb12@my.fsu.edu)

**Abstract.** The authors introduce a method for extracting weather and climate data from a historical plantation document. They demonstrate the method on a document from Shirley Plantation in Virginia (USA) covering the period 1816–1842. They show how the resulting data are organized into a spreadsheet that includes direct weather observations and information on various cultivars. They then give three examples showing how the data can be used for climate studies. The first example is a comparison of spring onset between the plantation era and the modern era. Modern median final spring freeze event (for the years 1943–2017) is occurring a week earlier than the historical median (for the years 1822–1839). The second analysis involves developing an index for mid-summer temperatures from the timing of first malaria-like symptoms in the plantation population each year. The median day when these symptoms would begin occurring in the modern period is a month and a half earlier than the median day they occurred in the historical period. The final example is a three–point temperature index generated from ordinal weather descriptions in the document. The authors suggest that this type of local weather information from historical archives, either direct from observations or indirect from phenophase timing, can be useful toward a more complete understanding of climates of the past.

## 1 Introduction

Weather data from historical periods is important for understanding climates of the past. Instrumental weather data go back only a few centuries but historical documents contain observations about the environment that, as a collection, can be used to understand the climate in previous centuries. In particular, plantation documents are an excellent source of useful observations about the local environment. These observations, when organized and collated, can enrich and extend our understanding of local climates further back into the past. In particular, the data introduced here does this for the southeastern United States. It complements other paleo and historical climate data for the early nineteenth century, providing a broader context in which to understand climate variability and climate change.

Plantation documents, a source for considerable historical research, are overlooked as a source for environmental research data, with a few notable exceptions (Burris et al., 2018; Mock, 2013; Dupigny-Giroux and Mock, 2009). Plantation documents often contain regular observations about the weather (Baptist, 2014; Mock et al., 2007; Phillips, 1918; Stampp, 1956). Where they have been used to study weather and climate, typically only a few ordinal data points are used in support of historical instrumental data (Dupigny-Giroux and Mock, 2009; Mock et al., 2007). Instrumental records are indeed available at certain times and places in these documents. For instance, Thomas Jefferson kept thermometer and barometer readings for his

Monticello Plantation for many years (Jefferson, 1776). But more typically, weather observations come in the form of ordinal observations. For instance, temperature might be described as "hot," "freezing," or "mild," to name a few (Table 6). Similar observations were made about wind conditions, cloud conditions, precipitation, and hydrology.

Plantation documents contain regular observations about crops, which can provide a proxy for the weather conditions under which they grow. For example, studies have used records of phenophases – stages in the life cycle – to study spring advancement or to reconstruct temperatures over past centuries (Aono and Kazui, 2008; Chuine et al., 2004; Primack and Miller-Rushing, 2012; Holopainen et al., 2013; White et al., 2018). These studies typically use one phenophase description from a single species. As a result, they are restricted to one data point per year. For instance, this might produce the number of days spring has advanced in a location over time, spring mean temperatures, or summer mean temperatures. In contrast, plantation document typically contain multiple phenophase observations for several crops each year, making them a particularly rich resource for proxy weather data.

The goal of this paper is to make plantation documents approachable to researchers interested in local weather conditions of the past by describing the process of turning historical documents into accessible scientific data. Here we use a document with daily entries from Shirley Plantation located in Charles City County, Virginia, USA covering the period of 1816–1842. The resulting database is then analyzed in three case studies. Replication of the method for analyzing documents available from other plantations will likely need some adjustments, but much of the organization structure should translate. Importantly, data of this kind covering a wider range of dates and other locations can be valuable for understanding our past climate for times and places that have minimal coverage from other sources.

This paper expands on earlier efforts to use a plantation document as a source for historical weather data and offers a structured method for turning a plantation document into usable data for climate research (Burris et al., 2018). Section 2 provides a brief history of extracting weather data from historical document. Section 3 describes how information from a historical document of Shirley Plantation is organized as a database. Section 4 discusses the agricultural variables that are included in the database. Section 5 describes the resulting database and illustrates how these data can be used for addressing scientific questions using case studies. Finally, section 6 provides a summary and some caveats and considerations for future studies.

## 2   Weather Data From the Historical Document

Historical documents have become an important source in climate research. The breadth of document types that have been used is staggering. As an example, researchers have used agricultural records, ship logs, port authority records, municipal records for harvest dates, newspapers, poetry, paintings, and financial reports, to just name a few. A wide array of observation types can be found in these sources. Researchers have been as creative in finding climate signals in documents as they have been in natural archives like ice cores and tree rings (White et al., 2018).

The geographical range of these sources are also diverse. Historical climatology is particularly well developed for China and Europe. By comparison, research into North America is less robust. The state of historical climatology for North America

is the result of a couple of factors. Compared to Eurasian documents, written records for North America started up recently. For instance, administrative records are available for China going back two thousand years, while records for the western hemisphere mostly start up after European contact. Few records prior to the sixteenth century survived. Those records that did exist prior to 1492 were systematically purged by the Aztec Empire in c. 1380, and again by the Spanish Spanish Empire during the sixteenth century (White et al., 2018).

As European Empires colonized more of the western hemisphere, records began to be produced for these locations. What is now the Southeastern United States, excluding Florida, was one of the last areas to see permanent European colonies. These colonies were also less centralized, both in regards to population and administration (White et al., 2018). This, combined with a dispersed economy based on slavery and agriculture, means that there is comparatively few sources to work with when studying this area (Burris et al., 2018; Nelson, 2007; White et al., 2018).

## 3 The Shirley Plantation Document

This section describes how information from a particularly rich Shirley Plantation document was organized into a database. Shirley Plantation is one of the oldest and largest plantations in the modern United States. It is located in Charles City County, Virginia, USA (Figure 1). Hill Carter inherited it in 1806 and started documenting events occasionally in a log book starting in 1816. He began keeping daily entries in 1820 that continued until his death in 1875 (data for 1840 is not in the database. The records exist for the year, but have not been extracted into the database). Hill Carter's son, Robert Randolph Carter (1825–1888), took over management of the plantation and continued documenting events until his death. The Shirley Plantation document refers to this single yet extensive log book.

Hill Carter's plantation document records many types of observations including, but not limited to, a weather diary, plant-phenology observations, ice-phenology observations, and the phenologies of other species that are responsive to climate variation. These other phenologies include fungus, nematodes, insects, and disease causing parasites. These often came in the form of crop pests, like wheat rust, cockle, and Hessian Fly, as well as well public health factors that we now know as *Anopheles* mosquitoes and *Plasmodium vivax*. Carter is a particularly good source because of his naval background. While serving as a young officer during the war of 1812, he would have learned to keep consistent and accurate records. While he did not immediately apply this skill set when he took over Shirley Plantation in 1816, by 1820, he was making consistent daily entries (White et al., 2018).

The primary crops grown on the Shirley Plantation during this time as noted in the logbook included wheat, corn, and clover. Several other species were grown intermittently (see Figure 2 for a full list of crops and the years they were mentioned in the document). When and how often crops showed up in the document varied. Staple crops like wheat and corn appeared every year, while other species like beans were rarely mentioned. The set of crops remained consistent over the period but it took a few years for Hill Carter to begin regular entries concerning them, resulting in the years 1816–1819 having very few entries. The first two years of consistent entries (1820–1821) were defined by infrequent observations. For instance, missing are entries of when the first and last freezes occurred. Beginning in 1822, Carter kept more consistent documentation. Entries became

more common until they were on a regular daily basis. The level of detail in the individual entries increased over this period (Figure 3).

Each year the number of observations increased throughout the growing season, culminating in the harvest of corn in the fall. This general cycle was disrupted by business trips to New York by Hill Carter. These trips were usually taken in October of each year, resulting in minimal entries for these periods. This can be seen in Figure 4, where the number of entries usually dipped in October. Hill Carter had the overseer make entries in his absence, with mixed results. Upon Hill Carter's return from these trips, he often expressed dissatisfaction about the how things were managed and documented by the overseer. After firing the overseer for this mismanagement in 1824, and then taking him back on, more consistent entries were kept during these trips, but these were still not up to the same standards. He did not make the journey in 1825, leaving that year's entries unbroken. After 1825, the trips resumed and the entries remained consistent for the rest of the years.

## 3.1 Observation Type

There is a range of observations within the document (Table 1). We group them into seven broad types. Weather observations have the most entries followed by Labor and Agriculture (Figure 5). Other categories include Operations, Human, Livestock, and Accounting. Typically, multiple observations are made in a single day's log entry. For this reason, there are multiple entries for a single day. A "type of observation" variable allowed for filtering the data based on activity (Figure 5).

### 3.1.1 Weather observations

The bulk of observations (13,461) made at Shirley Plantation were about the weather. Most days had at least one weather observation, even on Sundays, when there typically was no agricultural work done on the plantation. There were often two weather observations per day. One observation would involve temperature. The other would involve cloud conditions and precipitation. Most temperature and precipitation observations were ordinal and subject to the observer's definitions. Working with this type of data has been thoroughly researched elsewhere (White et al., 2018). As mentioned earlier, Carter's observations are notable for their consistency and reliability. Not only were the entries made daily, but they were generally made first thing in the morning with consistent language throughout the decades that are covered. In aggregate, these weather observations can be used to develop ordinal temperature indices and time series (White et al., 2018).

### 3.1.2 Agriculture observations

About one third of observations were considered agricultural. Most of the agricultural observations had to do with the work being done on the crops. This included activities like sowing, weeding, and harvesting. There were also discussions of problems encountered: for instance, weather–related problems like blow–overs and water damage from rain. There were also observations about pests including wheat rust, chinch bugs, and cockle (See Table 2 for definitions, and Table 3 for alternative terms and archaic spellings used in the document).

Direct observation of phenophases are rare. There are a few mentions of phases such as the day that a crop emerged, or when a field of grain started heading. Other phenophases can be deduced from agricultural activity. For instance, most entries do not explicitly say that the cotton crop had leafed. However, cotton crops would be weeded regularly in their early stages. This would continue until they leafed. At this point the leaves would shade out the weeds. Laborers, both free and enslaved, would be redirected, often to weeding corn. There are other ways to determine that the cotton had achieved the leafing stage. Mentions of weeding the cotton crop might end, or the entry may say that the laborers were now weeding the corn (Toman, 2002).

It is important note that this was a period of agricultural change and experimentation in Virginia, and Hill Carter participated heavily in these experiments (Nelson, 2007; Olmstead and Rhode, 2008b; White et al., 2018). Fortunately, he was also assiduous in documenting these efforts. He was careful to specify when he was fallowing and which fields he was doing it to. He experimented with several fertilizers and soil treatments to address soil erosion and degradation, and experimented with many cultivars. He documented when and where these experiments were conducted. He also noted when he tried new cultivars. Throughout his experiments, he used a few main cultivars every year. While the use of the various cultivars may have resulted in inhomogeneities in the plant-phenological records, there is a consistent reference variety that these can be compared to (White et al., 2018).

### 3.1.3 Labor and Operation Observations

While many entries are about weather and crops, there is also information on operations and labor conducted on the plantation. Such activities were performed before planting and after harvesting. Operations ranged from field preparation like plowing, to thrashing out and delivering grain. Labor covered activities like hauling and construction. While this may seem like a straightforward part of plantation life, plantations were a confluence of multiple agricultural traditions, coming from many regions and linguistic origins. The result is a patchwork of synonyms, meanings, and spellings, and terms came in archaic and modern forms.

Multiple terms are used to describe similar preparation activities. Each is typically associated with a particular crop. For instance, cutting, onioning, and picking out were terms for harvesting associated with grains, onions, and cotton, respectively. But picking out could also refer to things grown in patches like peas and pumpkins (spelt in the archaic form "pumpions" in the document). All of them refer to harvesting those crops. There were multiple spellings for some words as well. For Instance, "bauks," strips of land where corn was planted (Table 2), was alternatively spelled as "balks" and "baulks." Plaster was sometimes spelled as "plaister." Table 3 lists some of the more common terms with alternative synonyms or spellings. Table 2 contains definitions for some of these less common and archaic terms.

During this period, plantation owners experimented with various soil treatments. For instance, the enslaved laborers occasionally limed a field. This practice of applying crushed limestone before planting would make the soil less acidic. This could help plants grow more readily and combated soil degradation. Marl was produced by altering the chemical structure of lime by baking it. This made it more effective than lime, but fulfilled the same function. Plastering is a similar activity to liming, in which gypsum or some other plaster is spread on the field to alter the pH balance or replace depleted nutrients in the soil. Other

treatments were intended to improve the soil quality. Manuring was a common method. This referred to spreading manure on a field as a fertilizer. It was usually done just prior to sowing activities. Green manures and cover crops were also common. Peas and beans were often plowed under instead of being harvested in an effort to improve deteriorated fields. These preparatory activities are not directly related to phenology but they can provide useful information. For example, in the absence of detailed

documentation, these preparatory activities can inform the researcher that seeds have not been planted for that crop yet.

### 3.1.4   Additional Entries

Entries throughout rest of the document fall into the less common categories. These minor categories were grouped into Accounting, Human, and Livestock (Table 1). For example, accounts of how much grain was produced and sold can indicate how heavily impacted harvests were by weather conditions. Human health and activities were often directly impacted by the

weather. Observations of when people were sick were made regularly. This typically included how many people were sick and and what they were believed to be sick with. Illnesses, deaths, and other observations about the human population can be useful for showing how often conditions for endemic diseases like influenza and malaria occurred at the plantation. Non-agricultural labor activities can inform the entries about prevailing conditions, such as repairing flood and hurricane damage. While large animal phenology tends not to be directly affected by spring advancement or other aspects of climate change, secondary and

tertiary impacts like trophic mismatches can impact livestock and their phenophases (Post and Forchhammer, 2008). Health problems and deaths among the livestock from extreme heat also gives insight into weather conditions.

## 4   Agricultural Variables

This section discusses the agricultural variables that were put in the database. We cover the specific information contained in each entry. The species and cultivar variables provide information on the plant or animal being observed. Description provides

the account given in the plantation document. The categorical and numerical variables contain the specific information that can easily be used in quantitative analysis, such as how many bushels were harvested, or that the plantation was experiencing drought conditions. These translate into ordinal and interval data, respectively.

### 4.1   Species

Where the species was specified, the information was included. For non agricultural activity types, or when the species was not

specified, a "Not Available" (NA) value was entered. For instance, if the entry states that corn is being resown, then the species value would be "corn." Sometimes more than one species would be involved. The most common case of this was when a pest was effecting one of the crops. In these cases, the crop being grown would be put in the species variable. The pest species would be placed in the categorical variable. Figure 2 lists all species encountered in the Shirley Plantation document and the years that species was mentioned.

## 4.2 Cultivar

Specific cultivars of species were occasionally mentioned. Table 4 lists the cultivars mentioned within the document for the years 1816–842. Cultivar is part of the taxonomic nomenclature applied to plant species that have been artificially selected. The nomenclature goes from the more general to the specific, and is genus, species, subspecies, cultivar. Tracking the cultivar is especially useful for the nineteenth century. New cultivars were developed, experimented on, and further selected for specific traits. These efforts resulted in significant modification of the phenological and morphological traits in the crops. These modifications probably resulted in many of the crop yield gains seen in the antebellum period, as well as the expansion of the ranges of many crop species (Olmstead and Rhode, 2008a, b, 2018).

## 4.3 Description

The specific observation or description was entered in this variable. For instance, the specific phenophase being observed would go here. So if corn was observed to be tasseling on a certain day, the entry would read tasseling. If the document noted that the cotton was weeded for the last time and work began on weeding the corn, there would be two separate entries. There would be one entry for the end of the cotton weeding, and a separate entry for the beginning of the corn being weeded.

Types of activity for non-agricultural observations are part of this variable as well. Entries for agricultural operations were given more specificity. If some sort of soil treatment was being applied to a field, the type (such as marl, plaster, manure, etc.) would be entered here. This was also where the different type of weather observations were denoted. For instance, this variable would specify if the weather observation was about temperature, precipitation, or cloud cover. For health observations, the disease or cause of death was placed here.

The variables "Observation" and more general "Type of observation" make it easy to subset the descriptions. It is easy to just look at descriptions of agriculture activity, or to just look at when the human population was sick. The structure of the data makes it easy to input all the observations into a single database as the entries are processed. This makes the data entry more efficient while still being comprehensive, but easily reducible to a subset needed to study a particular phenomena.

This is an improvement on previous methodologies that have been more ad hoc (Burris et al., 2018). Instead of scouring the document for just the last spring freeze event or the flowering of a particular species, this method retains the robustness of the original document. This increases the usability of the data. Within one file, there is information on extreme weather events, phenology, public health, bioclimatology, and agronomy. Beyond the immediate concern of studying the past climate, the data has great interdisciplinary utility. It can be used in environmental and social history, economic research, epidemiology, and other disciplines. The initial investment of time is substantial, but the returns are worth it.

## 4.4 Categorical

The categorical or ordinal value of observations were placed here. This was most important for weather. There was usually some sort of descriptor accompanying the weather observation. For instance, if the document said that there was mist on a

certain day, the categorical value would be "mist." If an observation noted that it was "cold" on a certain day, it would be entered as "cold." This is also where freezes and frosts were entered.

This applied mainly to non-agricultural activity entries, but there were a few exceptions. When a crop was damaged or destroyed by pests, the nuisance species would be entered here. For instance, when a wheat crop was destroyed by rust, the "Observation" value would be pest, and the categorical value would be the type of pest, in this case rust. When an agricultural activity, such as sowing or weeding, would begin or end, that would be entered here as well. Units for numerical values were made available here.

## 4.5  Numerical Entries

Numerical values were occasionally available in the document. These values are entered in the database. If 200 bushels of wheat were sold after harvesting, "bushels" would be entered in the categorical variable, and "200" would go in the numerical variable. This was used to enter a variety of observations, including the number of people sick on a given day, thermometer readings, accounting values, etc.

There were intermittent instrumental readings from a thermometer starting in 1822. Temperatures were entered on an irregular basis until the instrument fell and shattered on February 23, 1824. It was not replaced for several years. Thermometer readings did not start up again until February 4, 1835. The document did not discuss the model of thermometer or where it was kept. While the units were not explicitly stated, summer temperatures were recorded as being as high as the nineties, clearly indicating that measurements were in degrees Fahrenheit rather than degrees Celsius.

These thermometer readings were entered in short spurts at different times of the year. When present, they were typically the first thing in that day's entry. The entries' consistent location in the daily log suggests that the instrumental reading was made first thing in the morning, when the daily temperature is usually at its lowest. The end result is a fairly homogeneous document of daily minimum temperatures during a few periods. The presence of the thermometer at Shirley Plantation, as opposed to being read about in an almanac or newspaper, was confirmed when it was noted that it fell to the ground and shattered (White et al., 2018).

There were limits to the precision of the instrument itself. Without knowing the details of its manufacture, it is not possible to tell how big of a measurement error there may or may not have been. The placement of the instrument, such as in the shade or direct sunlight, could also affect readings. There could be a systematic bias in how the observer was reading the instrument, such as if the mercury or alcohol in the instrument was between two ticks, the observer might always round it up or down. It is also possible for the observer to read the instrument differently from day to day, or for their method to change over time. The documenting of observations throughout the year was also inconsistent. The observer might only remember to document the temperature when it is especially cold or hot. They might also forget to make the entry if they are especially busy. This can be random, or systematic. During the calving season each year, when the observer is working long and irregular hours, they might forget to enter the temperature. They might also think to make an entry in response to extreme weather events when they would not have usually remembered to. There can also be more traditional inhomogeneities in the entries. If the instrument is moved

to a different location or height, a systematic bias can be introduced from suddenly being in direct sunlight or being closer to the ground. For a more detailed discussion of early instrumental records, see White et al. (2018).

## 4.6 Metadata

The database includes information on the time and place for each observation. This comprised information like the location of the plantation, date of observation, citation information and the name of the location. While most of these will not be directly used in the data analysis, they provide other functions. The location data can be used to geo-reference the database. Common names are useful for connecting the observations with mentions of the location in other document such as correspondences. Citation information allows for reproduction, confirmations of accuracy, and other forms of peer review. Location information will later enable areal interpolation of the data as the documents from additional locations are added to the database.

# 5 The Database

In this section we describe the resulting database and give three examples of the types of analyses that can be done with it. The first compares final spring freezes from the historical period to modern conditions to test for spring advancement. The second analysis looks at historical accounts of annual disease outbreaks of what is now thought to have been malaria. We use modern data to reconstruct when these outbreaks would occur, and compare the two sets of dates. A three–point temperature index is developed from the ordinal temperature observations.

## 5.1 Description

Here we provide some details about the database and then illustrate how it can be used. The Shirley Plantation documents were scoured for weather and agricultural information along the lines described in the previous section and arranged into a database. The result is 22,019 observations in the database. Table 5 shows a small portion of the database covering observations during the first week of July in 1838. Reading from left to right, each successive variable becomes more specific. After the date, the most general variable, "Observation Type", is given. This is followed the more specific category, "Observation". For instance, "agriculture" is one of the observation types. The agricultural observations for this week included "shelling", "harvesting", "problem", and "phenology". Wheat is the only species being worked on, and on four of the occasions it was specified that the cultivar of wheat being worked on was "purple straw".

We can see that on the first two days of the week, the wheat is harvested and shelled, often referred to as thrashing (See Table 3 for a variety of terms that referred to similar activity). Two problems with the crop were noted: that it was too ripe, and that part of the wheat had become tangled, making it more difficult to harvest. On the third day, there were more problems reported. A hurricane had gone through the area, causing the loss of some of the wheat crop. Additionally, some of the wheat that had been cut and left in rows (called windrows) had begun to open up and sprout. This was a result of the crop being over ripe and getting wet. For the rest of the week, the observations focus on harvesting the wheat crop, while continuing to note that it was too ripe.

There are also observations with the Observation type "Weather". Under this variable, ten of these were about temperature, with three of these also giving a thermometer reading. The other seven were ordinal observations, describing five of the days as "hot", two of the days as "cool", and one day as "warm". There were observations about cloud cover and precipitation as well. There were five "clear" days and two days with clouds and thunderstorms. The $6^{th}$ of July was described as windy. Rain was observed on the $1^{st}$ and $2^{nd}$ of July.

The database can be examined from various perspectives. For instance, the heat map in Figure 2 was constructed by using only database entries with the observation type "agriculture." The database closely reflects what was written in the historical document. If the document said that it rained and thundered, then two separate database entries would be entered. One for the rain, and one for the thunder, as is the case on July $1^{st}$, 1838 (Table 5). If something specific is required for analysis, such as days with cloud cover, the database can be subset to include only entries of "cloudy" and days with "precipitation."

## 5.2  Earlier spring onset

Here we illustrate how the database can be used for scientific studies. We start with investigating spring onset. Here we ask if the onset of spring occurring earlier now than it did during the study period. Specifically, given the warming climate due to greenhouse gases from industrialization, we expect that the annual dates of the last spring freeze over the period 1822–1839 (before modern industrialization) will occur later, on average, than the annual dates of the last spring freeze over the later period 1943–2017.

To examine this hypothesis we use the last date on which there was a mention of frost, freeze, or snow (after the winter) from the plantation document for each year. The start year of 1822 is the first year that mentioned freezes. We compare these dates with minimum temperature data from the nearest weather station. The weather data was retrieved from the National Oceanic and Atmospheric Administration's (NOAA) National Centers for Environmental Information's (NCEI) online repository. Daily Minimum temperatures from the Richmond International Airport weather station (GHCND:USW00013740) over period of January 1943 to July 2017 (NOAA, 2018). The station is located 20 kilometers to the Northwest of Shirley Plantation (See Figure 1 for a map of the area).

Dates are expressed as number of days into the year (day of year), with the first of January equaling one. It is possible that temperatures for the historical decade dropped below zero degrees Celsius without frost or snow being observed, so the given historical values are a more conservative measurement of the last freeze (could have been later) relative to the hypothesis. Figure 6 is a histogram of last spring freeze events in the instrumental period. The vertical black line represents the median value for the historical period. The historical database had a median end-of-freeze date of 107.5 (April 17 or 18). The modern median final spring freeze event date was 100 (April 10 or 11), or a difference of 7.5 days.

Because the historical last spring freeze data constituted a small sample size, a nonparametric one sided Wilcoxon (Mann-Whitney) signed-rank test was used to test the statistical significance of historical final spring freezes coming later in the year than in the modern instrumental period (Wilcoxon, 1945). The W value for the test was 797, with a $p$-value of 0.015. This indicates that the two groups' distributions differed in a statistically significant way, with modern final spring freezes occurring earlier than they did in the past. This indicates that spring onset is occurring earlier than in the historical period. The difference

in freeze dates is smaller than the one found in a previous study that only covered the historical period 1822-1828. In that study, there was a difference of 23 days, indicating much cooler temperatures in the 1820's compared to the 1830's (Burris et al., 2018). The example shows how the data might be useful for comparing the past with the present. It can be used to study how differing climate conditions have impacted crop survivability, yields, and how agricultural practices have changed in response to climate variation.

## 5.3   "Ague and Bilious Fever" first new infection date as indicator of mid summer temperature

Before germ theory and microscopes, malaria was often diagnosed as "ague and bilious fever" (See Tables 3 and 2 for related terms and their definitions). Figure 7 shows the period each year when "ague and bilious fever" symptoms were reported on Shirley Plantation. This covers the years 1823–1842, with 1839–1841 missing data. The earliest reports of symptoms occur in mid to late July, indicating especially warm years. Most years saw outbreaks begin in September. A few years only saw mentions of ague and fever in the first few days of October. These are often reports that there were very few or no cases that year. The season typically ends before Hill Carter left on his annual trips, resulting in complete documentation for most years. The parasite *Plasmodium* is responsible for the disease known as malaria. *Plasmodium vivax* is one of the mosquito borne pathogens that cause malaria. *Plasmodium falciparum* is another, more deadly, malaria causing parasite. *Plasmodium malariae* is a third, though less common *Plasmodium* pathogen (Guerra et al., 2008). This parasite has a complex life cycle and has specific environmental requirements that must be met for it to be able to infect human hosts. The beginning of the *Plasmodium* season, when symptoms emerge, the end of the *Plasmodium* season, marked by the last new case, and the length of the season between those two dates also offer important information about weather conditions.

An index of mid summer weather conditions can be constructed using historical public health data. The phenology of insects, and the parasites they carry, are sensitive to temperature (Guerra et al., 2008). Plant phenophases like flowering and harvest dates have been used to study changes in climate conditions (Aono and Kazui, 2008; Chuine et al., 2004; Primack and Miller-Rushing, 2012; White et al., 2018). These types of indices, models, and reconstructions can be applied to other temperature dependent phenologies (Guerra et al., 2008).

*Plasmodium* parasite development response to temperature can be modeled similarly to how crop phenology is, with degree days (Guerra et al., 2008), using the following formula.

$$DS = \frac{DD}{\bar{T} - 18} \tag{1}$$

Where DS is the number of days it will take for the parasite to mature (Duration of Sporogony), DD is the number of degree days required for parasite to mature, $\bar{T}$ is the daily mean temperature, and 18 is the minimal temperature threshold for degree days to accumulate for the species (Guerra et al., 2008). The sporogony of *Plasmodium vivax*, the type of malaria parasite common in the Tidewater region of Virginia during the nineteenth century, requires 105 degree days to accumulate before it becomes infectious (Gething et al., 2011). Both the *Anopheles* mosquito and *Plasmodium* parasite respond to temperature in their development (Guerra et al., 2008).

It is not necessary to diagnose specific cases of *Plasmodium* infection from the historical document. We know that it was endemic to the area. We know that the population was affected nearly every year. We know that the phenological constraints resulted in the infection season occurring in the same time period each year, just like wheat harvests or cherry blossoming (Rutman and Rutman, 1976; Merrens and Terry, 1984; Centers for Disease Control and Prevention, 2018; Guerra et al., 2008).

This produces the parameter for the new index: first new cases reported. This can be represented as the day of year. There are a series of events that must occur before the first symptoms are detected. The female *Anopheles* mosquito must become infected while taking a blood meal from a human. The mean daily temperature must be warm enough for the sporogony cycle to begin. The mean temperature during sporogony must be high enough to complete the cycle within the lifespan of the mosquito host. The mosquito host must transmit it to a human. Finally, the parasite must incubate in the human host for 12–18 days (for *P.*

*vivax* and *P. falciparum*) before symptoms appear (Guerra et al., 2008).

By the time symptoms appear, the daily mean temperature in the area has been above $18°$ Celsius ($64.4°$ Fahrenheit) for two to four weeks. This can be further examined if we know what species of *Plasmodium* are endemic to a region. For instance, *P. vivax* can survive in cooler conditions better than others. Its lower temperature threshold for development allows for it to complete its life cycle in regions that would be too cold for *P. falciparum*. As a result, this is the dominant species in temperate

regions like south eastern England and Tidewater Virginia (US) (Guerra et al., 2008; Mann, 2015; Rutman and Rutman, 1976).

While malaria has been eradicated in Virginia, we can estimate by using temperature data when the sporological cycle (*P. vivax*) would complete. For instances, if malaria were present in Virginia today, the *Anopheles* mosquito would become infectious after a sustained warm period. Warm periods are defined using accumulated degree days. A degree day in this case is defined as the difference between the daily mean temperature and 18 when the daily mean exceeds $18°$ C. A sustained warm

period is one in which the accumulated degree days over a 26-consecutive day period exceeds 105. Our interest is in the earliest sustained warm period date of the year where the date is the last day of the 26-day period.

Using the Richmond International Airport Weather Station temperature data, we compute the accumulated degree days above $18°$ C in rolling 26-day increments to determine the earliest sustained warm period each year over the period 1942–2017. We add 18 days to this date to account for incubation period (conservatively as the incubation might be faster) giving us an estimate

of when we might get the first cases of malaria (onset date).

We then used the historical and expected modern onset dates as indicators of summer temperature conditions. The median first date of symptoms being documented on Shirley for the period 1822–1838 was the $230^{th}$ day of the year. The median date for when symptoms would begin occurring in the modern period from 1942–2017 was the $185^{th}$ day of the year (See Figure 8). Median dates for malaria symptoms conditions occurred 45 days earlier in the modern period than they did in the historical

period. This indicates much warmer conditions in the modern period.

It is important to recognize that temperature is not the only limiting factor in the development of *Anopheles* and *Plasmodium*. For instance dry conditions can leave more water stagnating, providing ideal development conditions for the larva and pupa stages of mosquitoes. Large amounts of precipitation on the other hand can wash away water, making it difficult for mosquitoes to complete their life cycles (Guerra et al., 2008; Gething et al., 2011; White et al., 2018).

Human activity can also alter the life cycles of *Plasmodium* and *Anopheles*. Even factors like proximity of dwellings and work spaces to areas where *Anopheles* eggs and larva develop can impact infection rates and timings. The timing of labor activities can also influence this. For instance, people on Shirley would be much more likely to get infected while working on reclaimed swamp land. This part of the plantation had trouble dealing with standing water and was in close proximity with the remaining swamp land (Stampp et al., 1985). Without having controlled for these other environmental conditions, this analysis can only be suggestive of changes in climate (Gething et al., 2011; Guerra et al., 2008; McNeill, 2010).

Like the comparison of final spring freeze events in the first analysis, this allows a comparison to be made between ordinal data, commonly found in historical document, with numeric data from modern instruments. In this case, we compared mid summer temperature conditions across epochs. Where the first analysis allows us to compare temperatures by using observations of water undergoing a state change at $0°$ C, the analysis of malaria symptoms allows us to analyze summertime temperatures. The later method can be extended to any place and time where malaria was endemic and its effects documented.

## 5.4 Three-Point Temperature Index

A three-point temperature index was created for the 1820–1842 period. This style of temperature index was first developed by H. H. Lamb (White et al., 2018). It organizes ordinal descriptions from historical records onto a three point scale. Cold conditions are given a negative value, hot conditions are given a positive value, and neutral conditions were given the value. Christian Pfister expanded on the method to create a seven–point index. However, the seven–point method requires a second dataset to validate the indexed data by. Since this is not yet available for the study region, we will be using the three–point index here (White et al., 2018).

Ordinal descriptions of temperature were organized into three categories: hot, neutral, and cold. Table 6 shows how the categorical temperature values were broken down into the three–point index. There were 12 values assigned to hot, 13 values were assigned to neutral, and 29 were assigned to cold. A rolling mean was taken for each day using the values for the 30 days surrounding it (the day's value, as well as the previous 15 days, and the 15 days after it). Figure 9 shows the index covering 1820–1842. The rolling mean allows the data to be smoothed without losing its daily resolution. The index is in line with what we expected. The temperature index is at its highest during the summer months and lowest during the winter. The temperature index data is available at https://github.com/gdb12/Historical-bioclimatology.

The daily resolution of this data has some advantages. Most ordinal temperature indices developed from historical documents are at the seasonal or annual level. Daily data will allow future research to investigate day-to-day variability. It can also be used in conjunction with daily thermometer readings and a Pfister scale to reconstruct temperatures at the daily, weekly, or monthly temperatures. This can then be aggregated to seasonal and annual levels so it can be used in conjunction with other reconstructions (White et al., 2018).

## 6 Summary

We extracted historical environmental and weather data from a plantation document written over the period 1816–1842 at Shirley Plantation in Virginia, USA. We detailed the methods used to extract the data from the document and made the resulting index available as a dataset. The method provides a recipe for others interested in extracting data from similar historical documents. The database covers 25 years of plantation entries resulting in 22,019 entries in the database. The majority of the entries are ordinal observations of weather. Currently the database contains more than 2,000 entries about agriculture that can be used for phenological based studies of climate. There are also intermittent instrumental entries.

This database makes contributions in a few arenas. One use of historical climate records is in conjunction with paleoclimate records, adding daily resolution data to what is available from methods like dendrochronology (Druckenbrod et al., 2003; White et al., 2018). These records can also function as a rosetta stone. Just within the Shirley Plantation's record, there are thermometer readings, extensive plant, ice, insect, and parasite phenological records, and ordinal weather observations. Once this data is calibrated and validated, it can be used to extend our understanding using these different types of data from other documents. For instance, the ice-phenological information can be combined with observations of the same type from other sources, such as other plantation records, personal diaries, travel accounts, and ship logs (The James River is navigable up to Richmond Virginia). These records also allow us to study the phenology of important agricultural species prior to the intensive modification of these species over the past two hundred years through selective breeding and, more recently, genetic modification.

This data is also unusual for when it is being produced. By the mid eighteenth century, most ordinal weather diaries had abruptly ended in favor of instrumental observations from thermometers, barometers, and other newly developed scientific instrumentation. As a result, there is very little overlap between ordinal and instrumental weather data. Most sources for historical climate data in Europe made this transition within a very short time span. This same scientific turn can be seen in many North American sources, such as in the records of Thomas Jefferson and James Madison (Druckenbrod et al., 2003; Jefferson, 1776; White et al., 2018). With so little overlap to train in transfer functions, it is often difficult to calibrate and validate ordinal data (White et al., 2018). The Shirley Plantation document continues in the ordinal tradition more than a century after most other such documents end. This is concurrent with some of Hill Carter's own intermittent instrumental measurements and other such records that were produced in the area. As a result, this database represents an extraordinary opportunity to test calibration methodologies (White et al., 2018).

Using the database we compared historical and modern spring onset dates and showed that the median spring onset occurs a week earlier in the modern period. As the mean global temperature increases, measures of spring onset have been shown to occur earlier. These changes in temperature regimes can result in shifts in the phenology of plants and animals. Member species of an ecosystem can respond differently to these changes. Some plants will respond to spring advancement by flowering earlier in the year. Other plants are not responsive to these changes. The pollinators will have their own responses to the changing climate. The end result is disruptions to the flow of energy and resources. This has been shown to cause loss of plant populations in places like Walden pound in Massachusetts (Primack and Miller-Rushing, 2012). These trophic mismatches have

also increased the infant mortality rate among caribou, which are dependent on plants being at their most nutrient rich during calving season (Post and Forchhammer, 2008). Studies like this help measure the severity of spring advancement. This can in turn be used to predict where trophic mismatches are most common and have the greatest impact. This type of information is key to planning mitigation strategies for dealing with global climate change.

Using our database, we also showed how public health data can be used as an index for summer temperatures. Because of the temperature sensitivity of the *Plasmodium* parasite, the daily mean temperature in the area has to remain above 18° Celsius (64.4° Fahrenheit) for two to four weeks for the first new infections to begin. Thus the first appearance of new infections indicates how warm the mid summer period was each year. This in turn can be used as a proxy to study the inter-annual variability of temperature in the absence of instrumental records. A three–point temperature index was developed using the

data. Future research will expand this into a validated seven–point index.

There are additional considerations to keep in mind when using the database. Although we have already put approximately 3000 work hours into the database, we are not yet half way through the Shirley Plantation document. And there is an additional 25 years of related documents maintained by Hill Carter's son. There are also other similar documents from locations throughout Virginia and the Southeastern United States. It is also important to keep in mind how these document were kept.

The absence of a document does not ensure the absence of the event. Hill Carter would write down the weather conditions in the morning and update the entry with the work conducted throughout the day. Changes in weather throughout the day would often be recorded, but it is not certain that all such changes were documented. A common problem with historical documents is that events in the night would not be observed. Hill Carter often observed that there had been rain in the night (such details can be found in the narrative variable of the database, as seen in table 5). However, is not clear how often this occurred without

him observing and documenting it. The descriptions themselves are often subjective as well. A pleasant day in winter most probably has different temperature conditions than a summer day that was described as pleasant.

Further, how plantation documents were kept can vary across space and time. The way Americans kept these documents may have been very different from other former British colonies. What was deemed important to creditors could also influence this. If an area was more likely to be financed by British banks than American, then there could be more similarities between

American cotton plantations and British sugar plantations, than with tobacco plantations in the Upper South. Different imperial metropoles could have different expectations as well. The French Empire could have different standards and practices compared to the British, Spanish, or others. Further complicating and enmeshing these systems was the transfer of territory. What became the Louisiana purchase changed hands between the French and Spanish several times before becoming an American territory. This resulted in documents from this area being in French, Spanish, Creole, and English, and having influences from the

French, Spanish, British, and American systems (Baptist, 2014; Beckert, 2015; Tadman, 2000). While any analysis conducted using these documents needs to keep the limitations in mind, they do not negate the value of these documents.

Future work will focus on validating the data, expanding the database, and producing an in-depth analysis of the climate of Virginia. The most immediate concern will be data validation and calibration. We will test to see how well the data captures large scale climate phenomena like ENSO, tropical cyclone activity, and the aftermath of the Mount Tambora eruption. Cal-

ibration has proven to be more elusive. While there are a few thermometer observations in the data, they are too rare to be

used for calibration. None of the local newspapers published temperature records for this period either. However, once additional plantation records from other nearby plantations, such as Berkeley Plantation (6 miles southeast of Shirley Plantation) will allow for content analysis validation (Moodie and Catchpole, 1976). Thomas Jefferson's temperature observations offer another possible dataset for validation and calibration. During his tenure as ambassador to France, Jefferson confirmed that

his own thermometer was consistent with the official instrumental records in France, and maintained twice daily instrumental and ordinal weather observations during his four year residency there. this will allow us to calibrate North American records to its European contemporaries, and validate other North American temperature records. Jefferson's own plantation records, from his garden book, have already been extracted to a database by our team (Jefferson and Baron, 1987). Work has begun on extracting Jefferson's temperature records, including his time in Paris, France (Jefferson, 1776).

Additional years at Shirley Plantation will be added to the available data. Documents from additional plantations in Virginia will be analyzed as well. Temperatures will then be reconstructed using the response of various species to prevailing weather conditions. This can then be used to create a "phenochronology," similar to dendrochronology, but using historical phenological observations of plants similar to what Aono and Kazui (2008) have done with cherry blossoms, and what Chuine et al. (2004) have done with grape harvests. Because entries for several different species (Figure 2) are available in the database, they can

be combined to refine reconstructions and fill gaps in the document for individual species.

*Author contributions.* GB, JE, and RD designed the study; GB, JW, OL, SD extracted the data and built the database; GB and JE analyzed the data; GB wrote the initial draft; and all authors contributed to writing and editing.

*Competing interests.* The authors declare that they have no conflict of interest.

*Acknowledgements.* The authors would like to acknowledge the financial support From the Florida State University Robert B. Bradley

Library Research Grant, Graduate School Dissertation Research Grant, and the Undergraduate Research Opportunity Program (UROP) and material grant.

*Code and data availability.* The temperature index data and code for this project are available at https://github.com/gdb12/ Historical-bioclimatology

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

**Figure 1.** Map of Tidewater Virginia. Shirley Plantation is located along the James River, 20 kilometers (12 mi) southeast of the weather station at Richmond International Airport. The inset includes the political boundaries in Virginia during the period of 1816-1842.

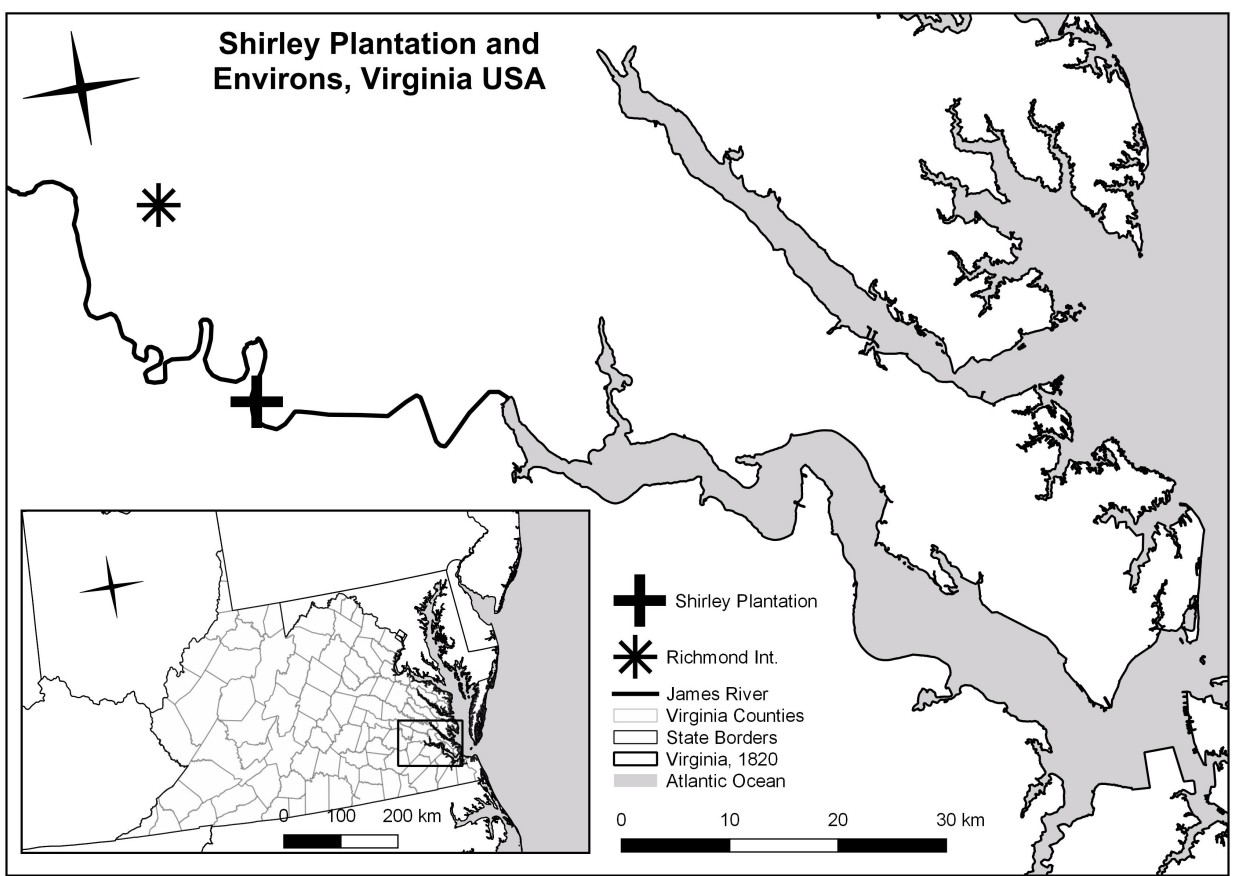

**Figure 2.** Heat map of species mentioned in the Shirley Plantation document. This heat map shows which years a crop was mentioned, and how many times it was mentioned that year.

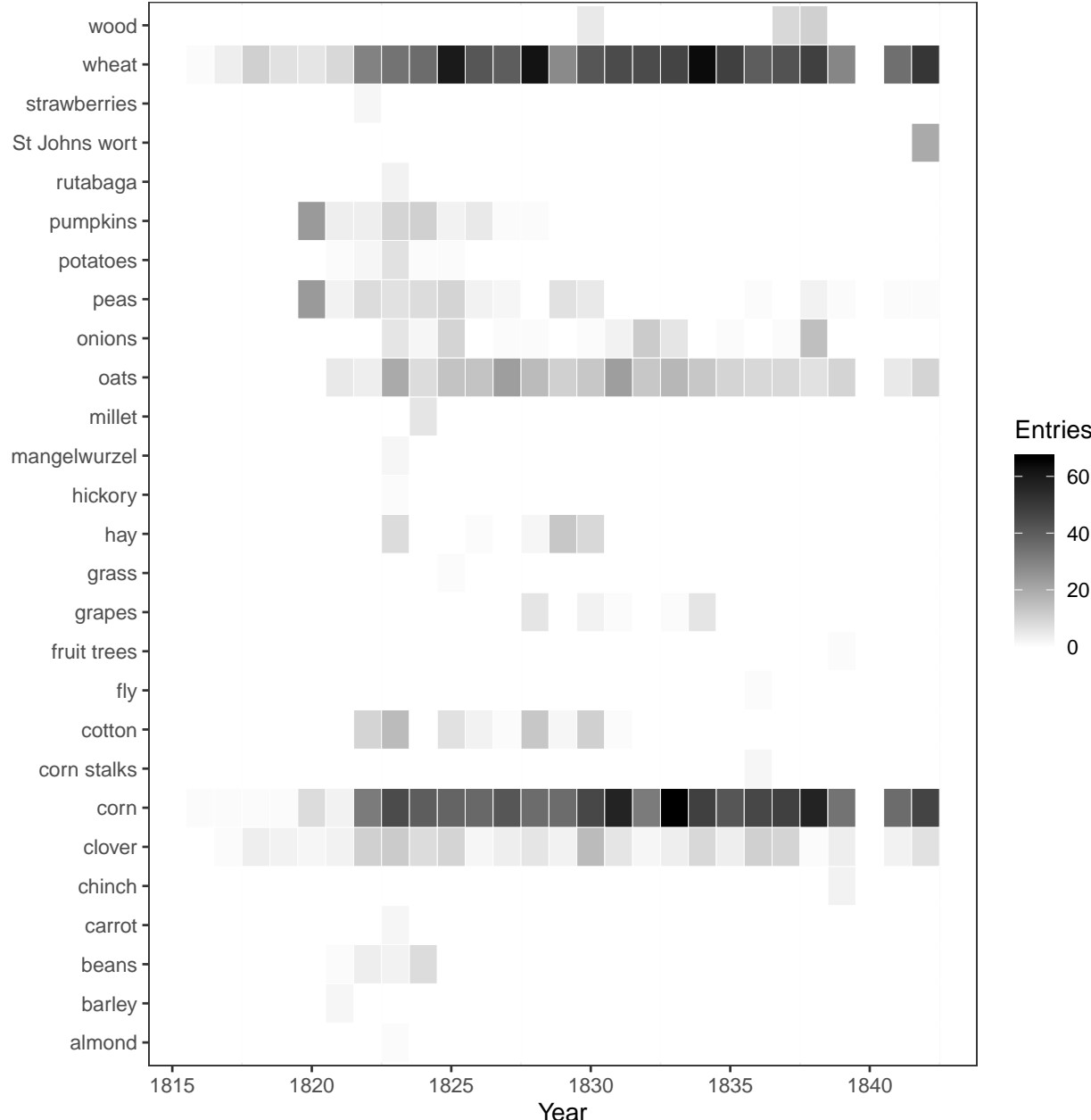

**Figure 3.** Number of database entries in non-overlapping 30-day periods extracted from the Shirley Plantation document.

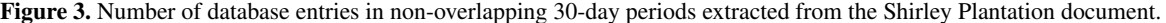

**Figure 4.** Number of database entries month. Activity on the plantation increased over the growing season, and then diminished over the fall and winter.

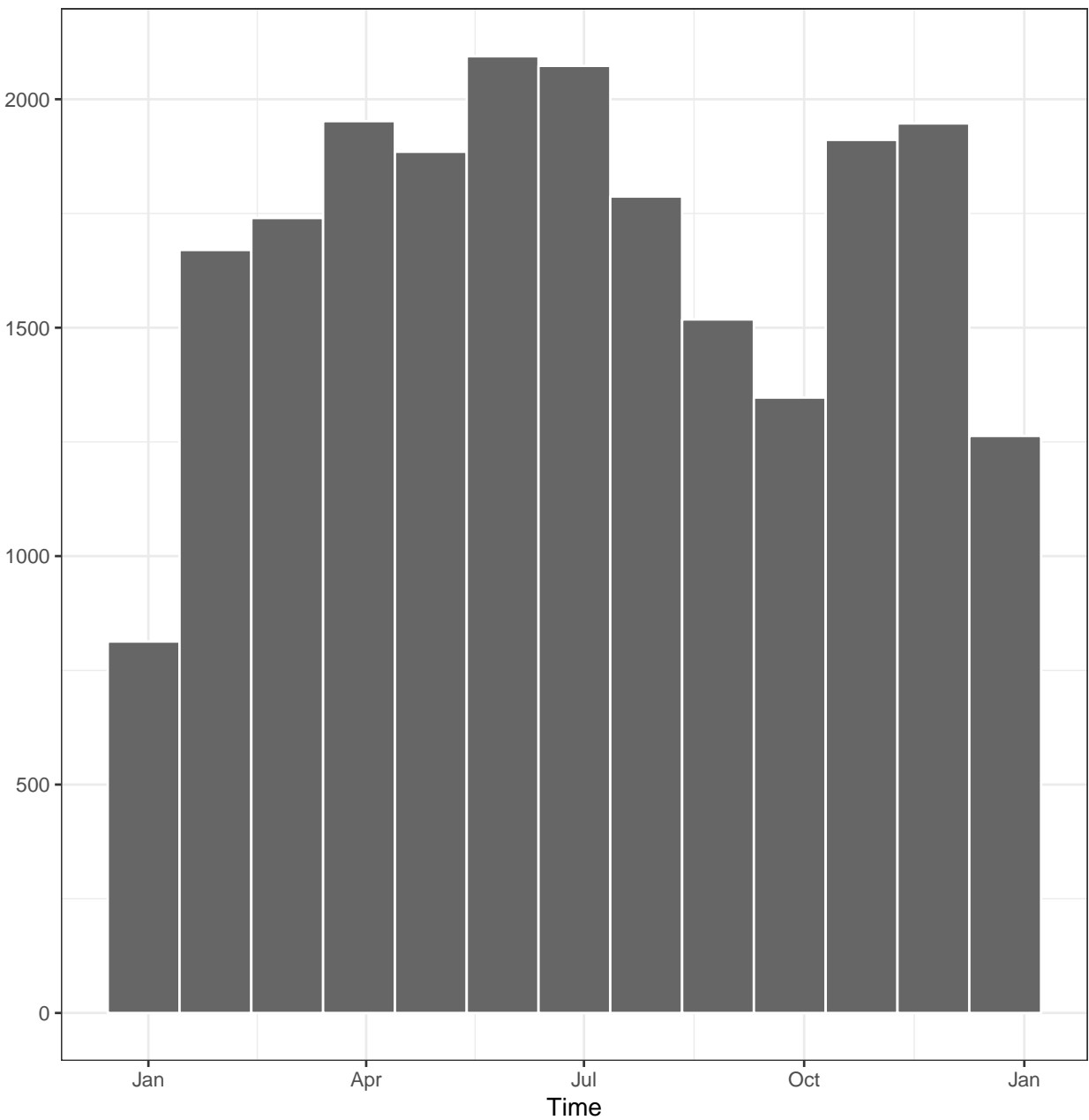

**Figure 5.** Entry type by observation category. The most common entry was about the weather, followed by agriculture, operations, and labor. Other categories had many fewer observations.

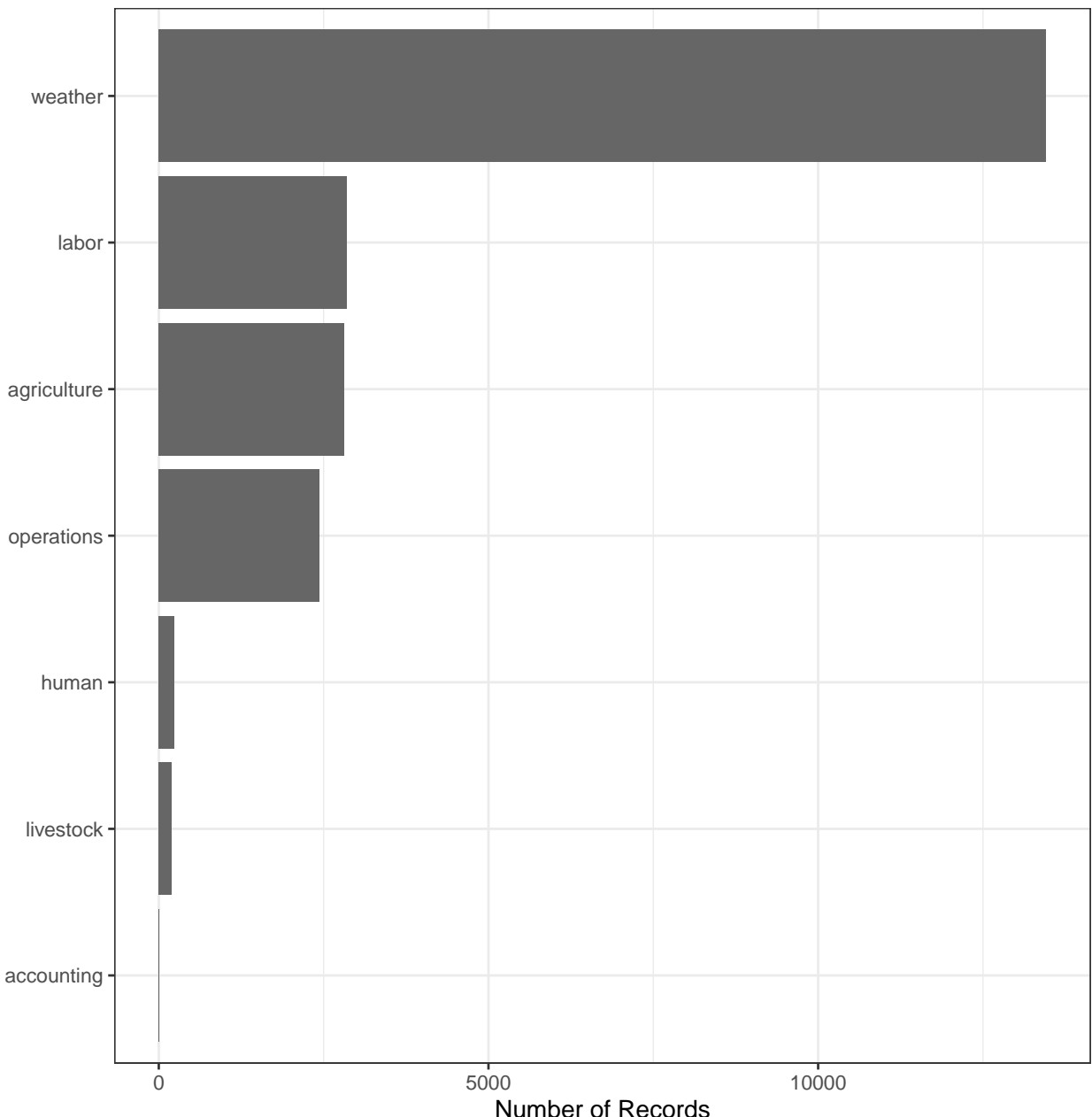

**Figure 6.** Histogram of last spring freeze days. This figure shows the distribution of last spring freeze days in Richmond, Virginia from 1942–2017. The black vertical line shows the median last spring freeze events in the data extracted from the Shirley Plantation document covering the period 1822–1838. There was one year where no freeze events were documented for the spring. This leaves 16 years in the historical set.

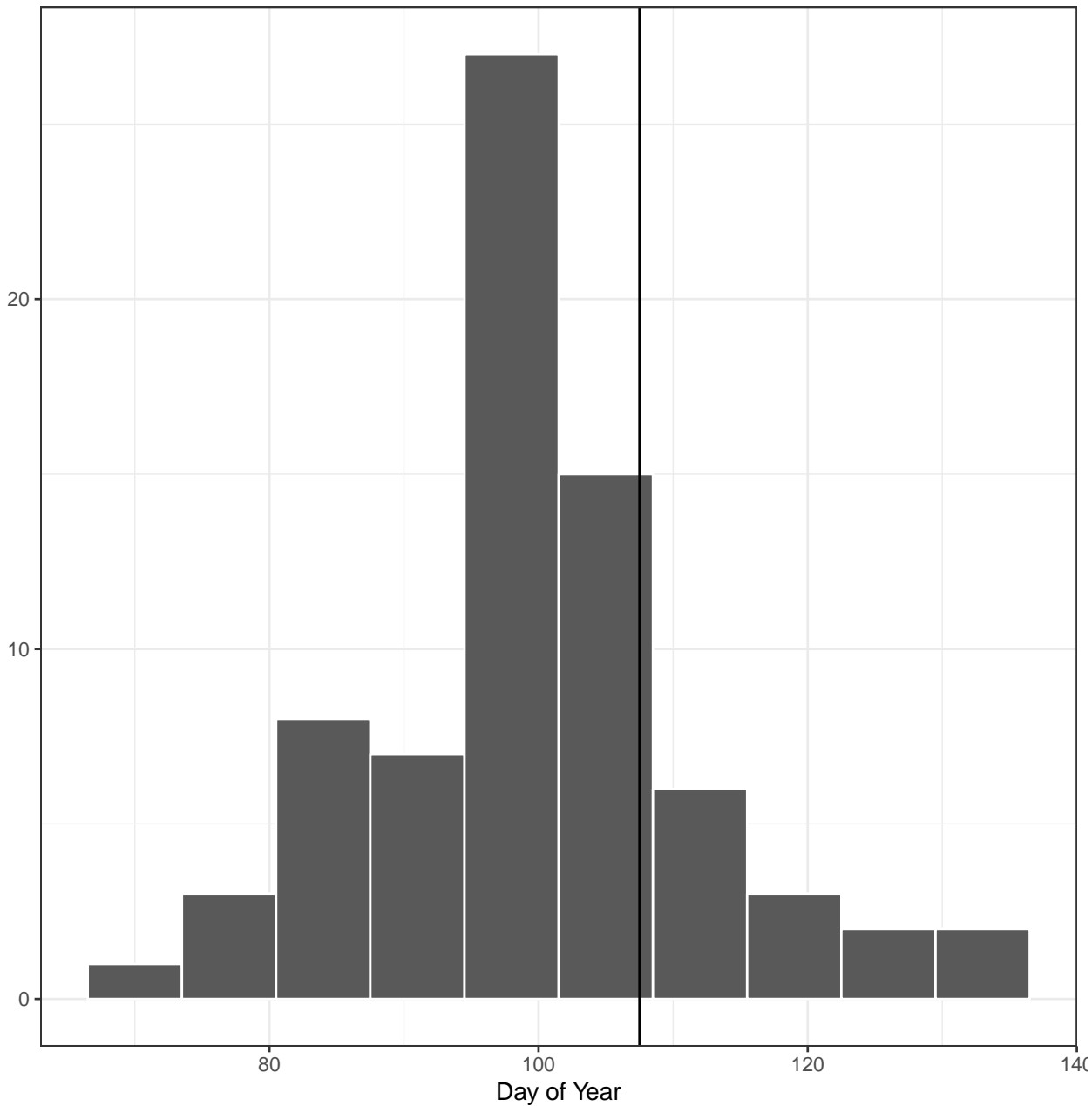

**Figure 7.** Reports of ague and fever in the Shirley Plantation document during the period 1823–1842. Data for 1839–841 has not been generated yet. There are two decades of data, minus the missing years. Onset of symptoms indicate that the mean daily temperatures had been above 18° Celsius (64.4° Fahrenheit) long enough for the sporological cycle of *Plasmodium* to complete in the host *Anopheles* mosquito. The green line indicates when the first case was reported. The red line indicates when the last report occurred.

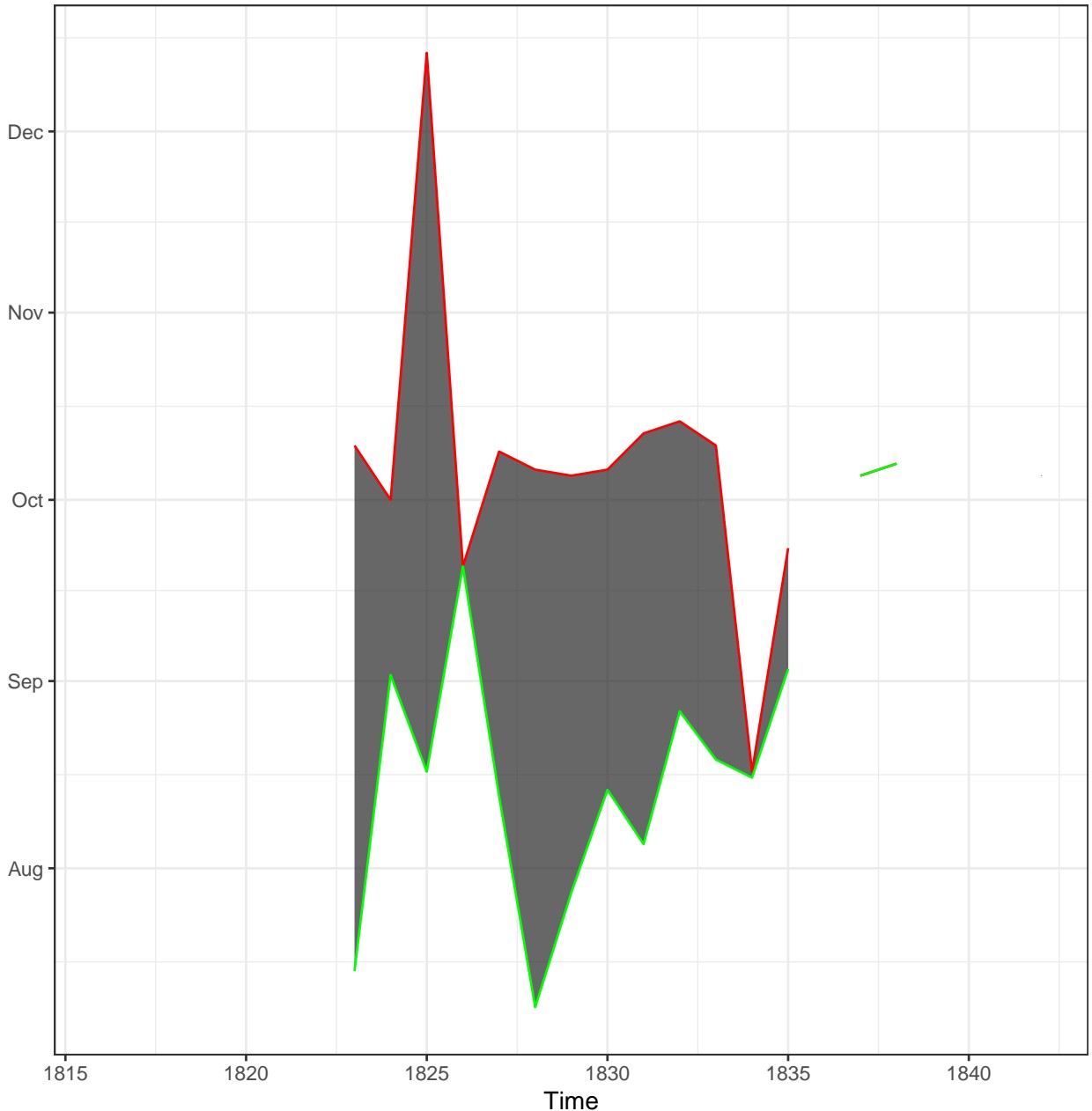

**Figure 8.** Histogram of expected first malaria symptoms in Richmond, Virginia using data from 1942–2017. The black vertical line shows the median first malaria symptom reports in the data extracted from the Shirley Plantation document covering the period 1822–1838.

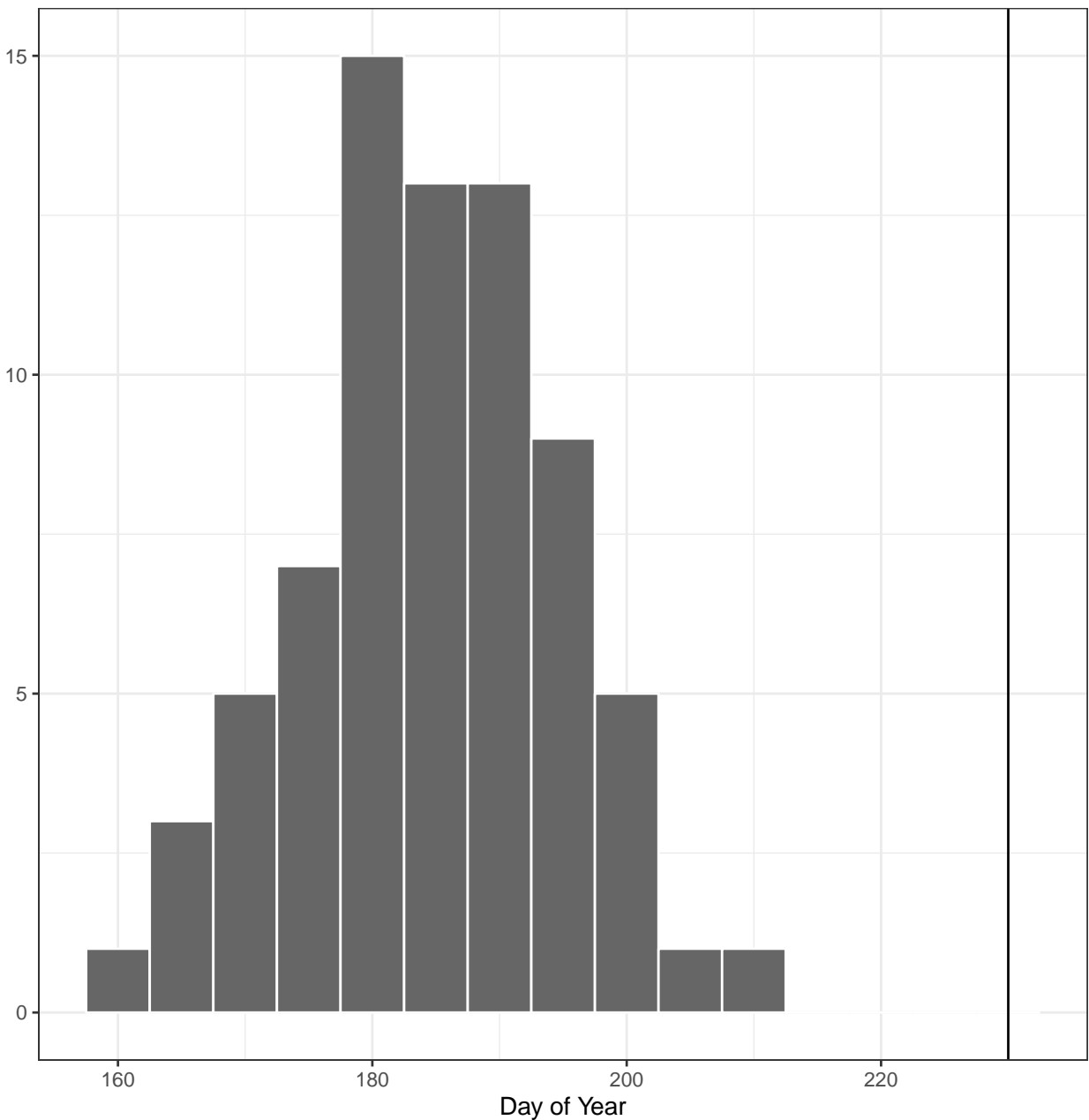

**Figure 9.** Three-point temperature index for the period 1816–1842. See text for discussion of how the index was constructed.

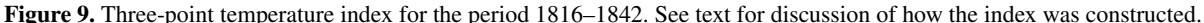

**Table 1.** Observation type and subcategories used in the Shirley Plantation document.

| Accounting | Agriculture | Human | Labor | Livestock | Operations | Weather |
|---|---|---|---|---|---|---|
| selling | experimenting | health | carpentry | fattening | burning | hydrology |
| | harvesting | holiday | constructing | feeding | clearing | precipitation |
| | pest | problem | delivering | fixing | dragging | problem |
| | phenology | runaway | general | health | drying | smokey |
| | problem | | hauling | managing | fallowing | temperature |
| | replanting | | preserving | moving | ginning | thundering |
| | sowing | | repairing | problem | hoeing | wind conditions |
| | thinning | | traveling | shearing | hydrology | |
| | weeding | | | | landscaping | |
| | | | | | penning | |
| | | | | | plowing | |
| | | | | | problem | |
| | | | | | shelling | |
| | | | | | treating | |

**Table 2.** Archaic terms encountered in the Shirley Plantation document.

| Term | Definition |
| --- | --- |
| ague | an archaic term for illness that is now typically diagnosed as malaria |
| bauks | strip of land where corn is planted |
| bilious fever | an archaic term for illness that is now typically diagnosed as malaria. Believed to be caused by an imbalances of the bile tempers of early western medicine |
| chinch | a pest insect species |
| cockle | a type of nematode that infects wheat |
| distemper | term for an illness believed to be caused by an imbalance of the four tempers |
| laying off | setting up drainage on a terraced field |
| listing | a partially mortared wall with turf on top |
| lodging | stalk collapse, esp. corn and wheat |
| marl | limestone baked in kilns that is then crushed and spread on fields to decrease the acidity of the soil, similar to liming and plastering |
| rust | a fungal crop pest |
| stob | fence post |
| windrowing | cutting a row of hay or small grain |

**Table 3.** Terms with similar meanings and spellings.

| Term | Alternate term 1 | Alternate term 2 | Alternate term 3 |
| --- | --- | --- | --- |
| ague and bilious fever | distemper | bilious fever | tertian fever |
| bauks | balk | baulk | |
| beveling | bavelling | bavel | laveling |
| chinch | cunch | conc | conch |
| cows | beef | beeves | bovines |
| harvesting | cutting | onioning | picking |
| hauling | moving | getting up | getting off |
| knocked down | blown over | prostrated | blown over |
| shelling | cobing | seeding | beating |

**Table 4.** Cultivars found in the Shirley Plantation document.

| Species | Cultivar | | | | | | | | |
|---------|----------|---|---|---|---|---|---|---|---|
| clover | pug | | | | | | | | |
| corn | hominy | gourdseed | shell | white | yellow | | | | |
| grapes | blue | orchard | | | | | | | |
| peas | partridge | cowpeas | galavant | | | | | | |
| wheat | white may | purple straw | red bearded | lloyd | red | turkey | rock | red may | domoville |

**Table 5.** Example of the database extracted from the Shirley Plantation document covering the period July 1–7, 1838.

| Date | Observation Type | Observation | Species | Cultivar | Narrative | Categorical | Numerical |
|------|------------------|-------------|---------|----------|-----------|-------------|-----------|
| 7/1/1838 | weather | cloud conditions | | | | clear | |
| 7/1/1838 | weather | temperature | | | | hot | |
| 7/1/1838 | weather | cloud conditions | | | | thundering | |
| 7/1/1838 | weather | precipitation | | | | rain | |
| 7/1/1838 | operations | shelling | wheat | | | shelling | |
| 7/2/1838 | weather | precipitation | | | | rain | |
| 7/2/1838 | weather | temperature | | | to excess | hot | |
| 7/2/1838 | agriculture | harvesting | wheat | purple straw | | harvesting | |
| 7/2/1838 | operations | shelling | wheat | | | shelling | |
| 7/2/1838 | agriculture | problem | wheat | | | ripe | |
| 7/2/1838 | agriculture | problem | wheat | | | tangled | |
| 7/3/1838 | weather | temperature | | | | Fahrenheit | 87 |
| 7/3/1838 | weather | cloud conditions | | | | clear | |
| 7/3/1838 | weather | temperature | | | to excess | hot | |
| 7/3/1838 | agriculture | phenology | wheat | | | opening | |
| 7/3/1838 | operations | shelling | wheat | | | shelling | |
| 7/3/1838 | operations | problem | wheat | | windrows | sprouting | |
| 7/3/1838 | agriculture | problem | wheat | | destroyed by hurricane | lost | |
| 7/4/1838 | weather | temperature | | | | Fahrenheit | 90 |
| 7/4/1838 | weather | cloud conditions | | | | clear | |
| 7/4/1838 | weather | temperature | | | hottest day this summer | hot | |
| 7/4/1838 | weather | cloud conditions | | | several | thundering | |
| 7/4/1838 | agriculture | harvesting | wheat | | | harvesting | |
| 7/4/1838 | operations | shelling | wheat | | | shelling | |
| 7/4/1838 | agriculture | problem | wheat | | | too ripe | |
| 7/5/1838 | weather | temperature | | | hotter than yesterday | hot | |
| 7/5/1838 | agriculture | harvesting | wheat | purple straw | | harvesting | |
| 7/5/1838 | weather | temperature | | | | Fahrenheit | 92 |
| 7/6/1838 | weather | cloud conditions | | | | cloudy | |
| 7/6/1838 | weather | temperature | | | great change | cool | |
| 7/6/1838 | weather | wind conditions | | | | windy | |
| 7/7/1838 | agriculture | harvesting | wheat | purple straw | | harvesting | |
| 7/7/1838 | weather | cloud conditions | | | | clear | |
| 7/7/1838 | weather | temperature | | | | warm | |
| 7/7/1838 | agriculture | harvesting | wheat | purple straw | | harvesting | |

**Table 6.** Temperature index Categories.

| Hot | Neutral | Cold |
| --- | --- | --- |
| exceptionaly hot | fair | below freezing |
| excessively hot | fine | cold |
| hot | mild | cool |
| hotest day this year | mild and pleasant | exceedingly cold |
| hotest morning this year | milder | excessively cold |
| indian summer | mildest winter ever known this far | excessively cold for season |
| sultry | moderate | first frost |
| very hot | nice | freeze |
| very warm | pleasant | frost |
| very warm for season | very mild | frost very slight |
| violently hot | warm | frozen |
| workers fainted from heat | warm and pleasant | ground frozen |
| | warmer | ice |
| | | light frost |
| | | river frozen |
| | | river hard frozen |
| | | river nearly frozen |
| | | second frost |
| | | severe frost |
| | | severe white frost |
| | | severest frost yet |
| | | smart frost |
| | | turned colder |
| | | turning cold |
| | | very cold |
| | | very cold violent change |
| | | very cool |
| | | white frost |
| | | wintery |