# Peer review of "Extracting Weather Information from a Plantation Document"

_Climate of the Past, 2018_

## Referee Comment (RC1) · Anonymous Referee #1 · 6 Dec 2018

This paper appears to make a solid contribution to the historical climatology of the southern United States by explaining and making accessible the extensive weatherand climate-related data of a historical document, the records of Stanley Plantation, Virginia. The paper explains the document's context, categories of observations, potential uses and weaknesses for climate and weather reconstruction; it provides two examples of how its records may be used to illustrate local climatic change between the period of observations (1816-42) and recent decades. It demonstrates the potential for further historical climatology based on plantation records. Within my range of expertise, I would judge the submission as nearly publishable as is, but I would recommend minor revisions:

-There are numerous minor grammatical and syntactical errors and passages that

could be made more concise and less repetitive.

-In section 3.1.1, it is important to note that historical climatologists working in other regional contexts (particularly early modern Europe and China) have long dealt with issues of the objectivity and potential for quantification of descriptive weather observations and have developed methods to address them such as monthly and seasonal indices (for an overview, see Christian Pfister and Sam White, "Evidence from the Archives of Societies: Personal Documentary Sources," in The Palgrave Handbook of Climate History, ed. Sam White, Christian Pfister, and Franz Mauelshagen (London: Palgrave Macmillan UK, 2018), 49–65, https://doi.org/10.1057/978-1-137-43020-5\_5). Therefore, with regard to these weather descriptions, it may be more helpful for the authors to discuss the particular strengths and weaknesses and the objectivity and subjectivity of these particular plantation records, rather than going over issues in personal weather descriptions in general. They may (although I wouldn't consider it necessary) also wish to propose a method for extracting objective and quantifiable information from the narrative descriptions in their records.

-While the article does mention changing agricultural practices at Stanley Plantation (p.5, lines 15-25), the authors could do more to contextualize and emphasize the magnitude of agricultural experimentation and changes in land use in Virginia at this time, under pressure from changing markets, declining soil fertility, and new notions of agricultural "improvement." This history is discussed in more detail in another close examination of long-term Virginia plantation records (although not for purposes of historical climatology): Lynn Nelson, Pharsalia: An Environmental Biography of a Southern Plantation, 1780-1880 (University of Georgia Press, 2007). I was surprised not to see this publication in the works cited, and I believe it would be helpful for this project. In this regard, or in section 4.2, the authors should also indicate (if possible) whether changing agricultural practices including the introduction of new cultivars appear to create artificial breaks or inhomogeneities in the timing of plant phenological observations.

-The authors mention crop pests in section 4.1: does the presence of any of these also
indicate specific climate conditions (similar to malaria, as discussed in the article)? If they do, it's worth mentioning; but if not, I don't recommend any changes. -The authors may save a few lines in the discussion of challenges in early thermometer readings in section 4.5 by citing to Dario Camuffo, "Evidence from the Archives of Societies: Early Instrumental Observations," in The Palgrave Handbook of Climate History, ed. Sam White, Christian Pfister, and Franz Mauelshagen (London: Palgrave Macmillan UK, 2018), 83–92, https://doi.org/10.1057/978-1-137-43020-5\_7 and Ingeborg Auer, "Analysis and Interpretation: Homogenization of Instrumental Data," in The Palgrave Handbook of Climate History, ed. Sam White, Christian Pfister, and Franz Mauelshagen (London: Palgrave Handbook of Climate History, ed. Sam White, Christian Pfister, and Franz Mauelshagen (London: Palgrave Macmillan UK, 2018), 99–105, https://doi.org/10.1057/978-1-137-43020-5\_9, which already summarize these issues. I offer this only as a suggestion.

-In general, I would like to see some verification of the accuracy and consistency of the climate information provided in the plantation documentation, or a suggestion how it might be verified. This verification could come from examination of its internal consistency: e.g., do variations in weather descriptions match variations in the timing of plant phenological observations? Or it could come from comparison with external information: e.g., does the Stanley Plantation document appear to consistently match other nearby records of climate or weather variability (e.g., stories about weather in a local newspaper, or descriptions of seasons in a local history) or well-observed extreme events that affected the region (e.g., the unusual winter of 1827-28 discussed in Cary J. Mock et al., "The Winter of 1827-1828 over Eastern North America: A Season of Extraordinary Climatic Anomalies, Societal Impacts, and False Spring," Climatic Change 83 (2007): 87-115, https://doi.org/10.1007/s10584-006-9126-2.)? In other words, even if the document does not provide a complete record of weather variability, is there a way to see if serves at least as a consistent high-pass filter for major variations and extremes? Even if the authors do not provide such verification here (and I do not regard it as essential for publication of this manuscript) could they suggest how it may be done in the case of this and other plantation records? That could be helpful for future research.

---

## Referee Comment (RC2) · Anonymous Referee #2 · 19 Dec 2018

There is much to like about this article. It explains how a source rarely used for quantitative environmental reconstructions - a plantation document - can in fact yield a wealth of remarkable data that shed light both on changes in the natural world and on social responses. The co-authors introduce a remarkable database - the product, apparently, of some 3,000 hours of work - that now has over 20,000 entries. They have the potential to make a substantial contribution to the disciplines of historical climatology and environmental history, one that certainly merits publication in a journal like Climate of the Past.

Yet their article, at present, does not adequately engage with prior scholarship in historical climatology; does not explain how its database improves existing climate reconstructions; and pushes some of its conclusions beyond what its data can sustain.

First, on engagement with prior scholarship: there is, of course, a very long tradition of European and Chinese historical climatology, and the authors should clearly explain how they are drawing from this tradition. In particular, it is striking to see no mention of Christian Pfister, whose work has made studies like this one possible. The authors attempt to describe scholarship that quantifies weather data in other documents, but their summaries a) do little justice to the sheer diversity of documents that historical climatologists have now deciphered, and b) scarcely engage with existing scholarship on the documents they mention. Any mention of ship logbook reconstructions, for example, should at the very least cite Dennis Wheeler and Dagomar Degroot.

Second, the authors should more carefully consider who is taking the observations they have databased, and why. Several additional sentences on the observers - ideally at the beginning of the article - would shed more light on how seriously we should take the observations. More importantly, the authors should clearly explain how their database can improve - or at least supplement - existing climate reconstructions that either draw from different documents or use entirely different sources (notably paleoclimatic proxy data and model hind-casting). To that end, the authors really should have included a reconstruction of temperature - on an ordinal scale - developed using their database. Does that reconstruction show what we expect? What does its (presumably) high resolution do for us?

Third, the section on malaria is poorly introduced. Terms and key relationships are referenced offhand before they are explained, and that will need some reworking. More importantly, it seems to me that the authors too directly link temperature to the first cases of malaria on the plantations. Of course, temperature has a major impact on malaria, but it is far from the only variable to consider. Plantations were artificial hotspots for malaria, and annual changes in first cases could well reflect human actions, decisions, and programs. Malaria is also very sensitive to rainfall, particularly alternating droughts and deluges (as John McNeill, among many others, has explained). At the very least, such complexity should be thoroughly acknowledged and articulated.

Finally, the article is, at present, rife with baffling typos and grammatical flaws - ones that should have been weeded out well before submission. It also contains some unintentionally comedic passages, such as: "summer temperatures were entered in the nineties, making it most likely that measurements were in degrees Fahrenheit rather than degrees Celcius" (and again, note the typo). Obviously these problems should be corrected for the revision.

In short: the article should be published, in my opinion, but only after major revisions.
* * *

---

## Author Comment (AC1) · 6 Feb 2019

article [utf8]inputenc graphicx

**Comment Response**

Gregory Burris

February 6, 2019

**1 Referee 1**

This paper appears to make a solid contribution to the historical climatology of the southern United States by explaining and making accessible the extensive weather and climate-related data of a historical document, the records of Stanley Plantation, Virginia. The paper explains the document's context, categories of observations, potential uses and weaknesses for climate and weather reconstruction; it provides two examples of how its records may be used to illustrate local climatic change between the period of observations (1816-42) and recent decades. It demonstrates the potential for further historical climatology based on plantation records.

**1.1 Author Response**

Thank you.

**1.2 Comments From Referees**

Within my range of expertise, I would judge the submission as nearly publishable as is, but I would recommend minor revisions: -There are numerous minor grammatical and syntactical errors and passages that could be made more concise and less repetitive.

**1.3 Author Response**

We have cleaned up the grammar and tightened up the prose. Please let us know if it remains a problem.

**1.4 Author's Changes in Manuscript**

[Changes were scattered throughout the manuscript]

**1.5 Comments From Referees**

-In section 3.1.1, it is important to note that historical climatologists working in other regional contexts (particularly early modern Europe and China) have long dealt with issues of the objectivity and potential for quantification of descriptive weather observations and have developed methods to address them such as monthly and seasonal indices (for an overview, see Christian Pfister and Sam White, "Evidence from the Archives of Societies: Personal Documentary Sources," in The Palgrave Handbook of Climate History, ed. Sam White, Christian Pfister, and Franz Mauelshagen (London: Palgrave Macmillan UK, 2018), 49–65, $https://doi.org/10.1057/978-1-137-43020-5_5$).

Therefore, with regard to these weather descriptions, it may be more helpful for the

authors to discuss the particular strengths and weaknesses and the objectivity and subjectivity of these particular plantation records, rather than going over issues in personal weather descriptions in general.

**1.6 Author Response**

Thank you for bringing the Palgrave Handbook to our attention. It was invaluable for improving the manuscript. We now spend less time on the weather descriptions and point to White et al. (2018) for further reading. We focus more on the particulars of the document we are working on.

**1.7 Author's Changes in Manuscript**

Carter is a particularly good source because of his naval background. While serving as a young officer during the war of 1812, he would have learned to keep consistent and accurate records. While he did not immediately apply this skill set when he took over Shirley Plantation in 1816, by 1820, he was making consistent daily entries (White et al., 2018).

[...]

The bulk of observations (13,461) made at Shirley Plantation were about the weather. Most days had at least one weather observation, even on Sundays, when there typically was no agricultural work done on the plantation. There were often two weather observations per day. One observation would involve temperature. The other would involve cloud conditions and precipitation. Most temperature and precipitation observations were ordinal and subject to the observer's definitions. Working with this type of data has been thoroughly researched elsewhere (White et al., 2018). As mentioned earlier, Carter's observations are notable for their consistency and reliability. Not only

were the entries made daily, but they were generally made first thing in the morning with consistent language throughout the decades that are covered. In aggregate, these weather observations can be used to develop ordinal temperature indices and time series (White et al., 2018).

**1.8 Comments From Referees**

They may (although I wouldn't consider it necessary) also wish to propose a method for extracting objective and quantifiable information from the narrative descriptions in their records.

**1.9 Author Response**

Referee 2 made a similar suggestion, recommending that we construct a temperature index from the data. White et al. (2018) was very useful here as well. Because we have not performed the data validation expected for a seven–point "Pfister scale" temperature index, we constructed a three–point index modeled after the one designed by H. H. Lamb (also discussed in White et al. (2018)).

**1.10 Author's Changes in Manuscript**

A three-point temperature index was created for the 1820–1842 period. This style of temperature index was first developed by H. H. Lamb (White et al., 2018). It organizes ordinal descriptions from historical records onto a three point scale. Cold conditions are given a negative value, hot conditions are given a positive value, and neutral conditions were given the value. Christian Pfister expanded on the method to create a seven–point index. However, the seven–point method requires a second dataset to validate the indexed data by. Since this is not yet available for the study region, we will be using
the three–point index here (White et al., 2018). This, combined with the rest of section 5 should accomplish this task.

Ordinal descriptions of temperature were organized into three categories: hot, neutral, and cold. Table 1 shows how the categorical temperature values were broken down into the three–point index. There were 12 values assigned to hot, 13 values were assigned to neutral, and 29 were assigned to cold. A rolling mean was taken for each day using the values for the 30 days surrounding it (the day's value, as well as the previous 15 days, and the 15 days after it). Figure 1 shows the index covering 1820–1842. The rolling mean allows the data to be smoothed without losing its daily resolution. The index is inline with with what we expected. The temperature index is at its highest during the summer months and lowest during the winter. The temperature index data is available at https://github.com/gdb12/Historical-bioclimatology.

1.11 Comments From Referees

-While the article does mention changing agricultural practices at Stanley Plantation (p.5, lines 15-25), the authors could do more to contextualize and emphasize the magnitude of agricultural experimentation and changes in land use in Virginia at this time, under pressure from changing markets, declining soil fertility, and new notions of agricultural "improvement." This history is discussed in more detail in another close examination of long-term Virginia plantation records (although not for purposes of historical climatology): Lynn Nelson, Pharsalia: An Environmental Biography of a Southern Plantation, 1780-1880 (University of Georgia Press, 2007). I was surprised not to see this publication in the works cited, and I believe it would be helpful for this project.

In this regard, or in section 4.2, the authors should also indicate (if possible) whether changing agricultural practices including the introduction of new cultivars appear to create artificial breaks or inhomogeneities in the timing of plant phenological observations.

[Figure]

**1.12 Author Response**

There was lot of experimentation on Shirley Plantation, including, soil treatments and new cultivars. Fortunately, these were well documented, and there were a few main cultivars that were always grown. These main cultivars can be used to create a homogeneous record, as well as allowing us to test for inhomogeneities in the expiremetal cultivars. The result is a consistent and homogeneous core of data, with experimentation being supplemental to this. The discussion was expanded to include this, and references to Nelson were added. We also added a discussion of the experimentation and changes in crops during this period to ensure that readers are aware of the possible complications.

**1.13 Author's Changes in Manuscript**

It is important note that this was a period of agricultural change and experimentation in Virginia, and Hill Carter participated heavily in these experiments (Nelson, 2007; Olmstead and Rhode, 2008b; White et al., 2018). Fortunately, he was also assiduous in documenting these efforts. He was careful to specify when he was fallowing and which fields he was doing it to. He experimented with several fertilizers and soil treatments to address soil erosion and degradation, and experimented with many cultivars. He documented when and where these experiments were conducted. He also noted when he tried new cultivars. Throughout his experiments, he used a few main cultivars every year. While the use of the various cultivars may have resulted in inhomogeneities in the plant-phenological records, there is a consistent reference variety that these can be compared to (White et al., 2018).

[...]

Tracking the cultivar is especially useful for the nineteenth century. New cultivars were developed, experimented on, and further selected for specific traits. These efforts

resulted in significant modification of the phenological and morphological traits in the crops. These modifications probably resulted in many of the crop yield gains seen in the antebellum period, as well as the expansion of the ranges of many crop species (Olmstead and Rhode, 2008a, b, 2018).

**1.14   Comments From Referees**

-The authors mention crop pests in section 4.1: does the presence of any of these also indicate specific climate conditions (similar to malaria, as discussed in the article)? If they do, it's worth mentioning; but if not, I don't recommend any changes.

**1.15   Author Response**

There are several types of pests that are sensitive to climate variation. We included a discussion of these in the new discussion in section 3. There is potential here, but it will take additional research to realize. These are now included in the discussion of the types of data available in the document we are working on.

**1.16   Author's Changes in Manuscript**

Hill Carter's plantation document records many types of observations including, but not limited to, a weather diary, plant-phenology observations, ice-phenology observations, and the phenologies of other species that are responsive to climate variation. These other phenologies include fungus, nematodes, insects, and disease causing parasites. These often came in the form of crop pests, like wheat rust, cockle, and Hessian Fly, as well as well public health factors that we now know as *Anopheles* mosquitoes and *Plasmodium vivax*.

**1.17 Comments From Referees**

-The authors may save a few lines in the discussion of challenges in early thermometer readings in section 4.5 by citing to Dario Camuffo, "Evidence from the Archives of Societies: Early Instrumental Observations," in The Palgrave Handbook of Climate History, ed. Sam White, Christian Pfister, and Franz Mauelshagen (London: Palgrave Macmillan UK, 2018), 83–92, $https : //doi.org/10.1057/978 - 1 - 137 - 43020 - 5_7$ and Ingeborg Auer, "Analysis and Interpretation: Homogenization of Instrumental Data," in The Palgrave Handbook of Climate History, ed. Sam White, Christian Pfister, and Franz Mauelshagen (London: Palgrave Macmillan UK, 2018), 99–105, $https : //doi.org/10.1057/978 - 1 - 137 - 43020 - 5_9$, which already summarize these issues. I offer this only as a suggestion.

**1.18 Author Response**

this section was shortened and reference given to the Palgrave Handbook. The section now focusses on the particulars of Shirley Plantation's instrumental records.

**1.19 Author's Changes in Manuscript**

There were limits to the precision of the instrument itself. Without knowing the details of its manufacture, it is not possible to tell how big of a measurement error there may or may not have been. The placement of the instrument, such as in the shade or direct sunlight, could also affect readings. There could be a systematic bias in how the observer was reading the instrument, such as if the mercury or alcohol in the instrument was between two ticks, the observer might always round it up or down. It is also possible for the observer to read the instrument differently from day to day, or for their method to change over time. The documenting of observations throughout the year

was also inconsistent. The observer might only remember to document the temperature when it is especially cold or hot. They might also forget to make the entry if they are especially busy. This can be random, or systematic. During the calving season each year, when the observer is working long and irregular hours, they might forget to enter the temperature. They might also think to make an entry in response to extreme weather events when they would not have usually remembered to. There can also be more traditional inhomogeneities in the entries. If the instrument is moved to a different location or height, a systematic bias can be introduced from suddenly being in direct sunlight or being closer to the ground. For a more detailed discussion of early instrumental records, see White et al. (2018).

1.20   Comments From Referees

-In general, I would like to see some verification of the accuracy and consistency of the climate information provided in the plantation documentation, or a suggestion how it might be verified. This verification could come from examination of its internal consistency: e.g., do variations in weather descriptions match variations in the timing of plant phenological observations? Or it could come from comparison with external information: e.g., does the Stanley Plantation document appear to consistently match other nearby records of climate or weather variability (e.g., stories about weather in a local newspaper, or descriptions of seasons in a local history) or well-observed extreme events that affected the region (e.g., the unusual winter of 1827-28 discussed in Cary J. Mock et al., "The Winter of 1827–1828 over Eastern North America: A Season of Extraordinary Climatic Anomalies, Societal Impacts, and False Spring," Climatic Change 83 (2007): 87–115, https://doi.org/10.1007/s10584-006-9126-2.)?  In other words, even if the document does not provide a complete record of weather variability, is there a way to see if serves at least as a consistent high-pass filter for major variations and extremes? Even if the authors do not provide such verification here (and I do not regard it as essential for publication of this manuscript) could they suggest how it

may be done in the case of this and other plantation records? That could be helpful for future research.

**1.21 Author Response**

We feel that data validation is the topic of my next paper and is a subject large enough for its own publication. We added a section on this to our section on future work. We discuss our plans for data calibration and validation. We intend to test to see if large climate phenomena like ENSO is detectable in the data. Once an appropriate dataset is found, we will also calibrate and validate more sophisticated calibrations.

**1.22 Author's Changes in Manuscript**

Future work will focus on validating the data, expanding the database, and producing an in-depth analysis of the climate of Virginia. The most immediate concern will be data validation and calibration. We will test to see how well the data captures large scale climate phenomena like ENSO, tropical cyclone activity, and the aftermath of the Mount Tambora eruption. Calibration has proven to be more elusive. While there are a few thermometer observations in the data, they are too rare to be used for calibration. None of the local newspapers published temperature records for this period either. However, once additional plantation records from other nearby plantations, such as Berkeley Plantation (6 miles southeast of Shirley Plantation) will allow for content analysis validation (Moodie and Catchpole, 1976). Thomas Jefferson's temperature observations offer another possible dataset for validation and calibration. During his tenure as ambassador to France, Jefferson confirmed that his own thermometer was consistent with the official instrumental records in France, and maintained twice daily instrumental and ordinal weather observations during his four year residency there. this will allow us to calibrate North American records to its European contemporaries,

and validate other North American temperature records. Jefferson's own plantation records, from his garden book, have already been extracted to a database by our team (Jefferson and Baron, 1987. Work has begun on extracting Jefferson's temperature records, including his time in Paris, France (Jefferson, 1776).

**2 New Figure and table**

**Fig. 1.** Three-point temperature index for the period 1816–1842. See text for discussion of how the index was constructed.

indexdaily.pdf

**Table 1.** Temperature index Categories.

| Hot | Neutral | Cold |
|---|---|---|
| exceptionaly hot | fair | below freezing |
| excessively hot | fine | cold |
| hot | mild | cool |
| hotest day this year | mild and pleasant | exceedingly cold |
| hotest morning this year | milder | excessively cold |
| indian summer | mildest winter ever known this far | excessively cold for season |
| sultry | moderate | first frost |
| very hot | nice | freeze |
| very warm | pleasant | frost |
| very warm for season | very mild | frost very slight |
| violently hot | warm | frozen |
| workers fainted from heat | warm and pleasant | ground frozen |
| | warmer | ice |
| | | light frost |
| | | river frozen |
| | | river hard frozen |
| | | river nearly frozen |
| | | second frost |
| | | severe frost |
| | | severe white frost |
| | | severest frost yet |
| | | smart frost |
| | | turned colder |
| | | turning cold |
| | | very cold |
| | | very cold violent change |
| | | very cool |
| | | white frost |
| | | wintery |

---

## Author Comment (AC2) · 6 Feb 2019

article [utf8]inputenc graphicx

[Figure]

**Comment Response**

Gregory Burris

January 2019

**1 Referee 1**

This paper appears to make a solid contribution to the historical climatology of the southern United States by explaining and making accessible the extensive weather and climate-related data of a historical document, the records of Stanley Plantation, Virginia. The paper explains the document's context, categories of observations, potential uses and weaknesses for climate and weather reconstruction; it provides two examples of how its records may be used to illustrate local climatic change between the period of observations (1816-42) and recent decades. It demonstrates the potential for further historical climatology based on plantation records.

**1.1 Author Response**

Thank you.

**1.2   Comments From Referees**

Within my range of expertise, I would judge the submission as nearly publishable as is, but I would recommend minor revisions: -There are numerous minor grammatical and syntactical errors and passages that could be made more concise and less repetitive.

**1.3   Author Response**

We have cleaned up the grammar and tightened up the prose. Please let us know if it remains a problem.

**1.4   Author's Changes in Manuscript**

[Changes were scattered throughout the manuscript]

**1.5   Comments From Referees**

-In section 3.1.1, it is important to note that historical climatologists working in other regional contexts (particularly early modern Europe and China) have long dealt with issues of the objectivity and potential for quantification of descriptive weather obser- vations and have developed methods to address them such as monthly and seasonal indices (for an overview, see Christian Pfister and Sam White, "Evidence from the Archives of Societies: Personal Documentary Sources," in The Palgrave Handbook of Climate History, ed. Sam White, Christian Pfister, and Franz Mauelshagen (London: Palgrave Macmillan UK, 2018), 49–65, $https://doi.org/10.1057/978-1-137-43020-5_5$).

Therefore, with regard to these weather descriptions, it may be more helpful for the

authors to discuss the particular strengths and weaknesses and the objectivity and subjectivity of these particular plantation records, rather than going over issues in personal weather descriptions in general.

**1.6 Author Response**

Thank you for bringing the Palgrave Handbook to our attention. It was invaluable for improving the manuscript. We now spend less time on the weather descriptions and point to White et al. (2018) for further reading. We focus more on the particulars of the document we are working on.

**1.7 Author's Changes in Manuscript**

Carter is a particularly good source because of his naval background. While serving as a young officer during the war of 1812, he would have learned to keep consistent and accurate records. While he did not immediately apply this skill set when he took over Shirley Plantation in 1816, by 1820, he was making consistent daily entries (White et al., 2018).

[...]

The bulk of observations (13,461) made at Shirley Plantation were about the weather. Most days had at least one weather observation, even on Sundays, when there typically was no agricultural work done on the plantation. There were often two weather observations per day. One observation would involve temperature. The other would involve cloud conditions and precipitation. Most temperature and precipitation observations were ordinal and subject to the observer's definitions. Working with this type of data has been thoroughly researched elsewhere (White et al., 2018). As mentioned earlier, Carter's observations are notable for their consistency and reliability. Not only

were the entries made daily, but they were generally made first thing in the morning with consistent language throughout the decades that are covered. In aggregate, these weather observations can be used to develop ordinal temperature indices and time series (White et al., 2018).

**1.8   Comments From Referees**

They may (although I wouldn't consider it necessary) also wish to propose a method for extracting objective and quantifiable information from the narrative descriptions in their records.

**1.9   Author Response**

Referee 2 made a similar suggestion, recommending that we construct a temperature index from the data. White et al. (2018) was very useful here as well. Because we have not performed the data validation expected for a seven–point "Pfister scale" temperature index, we constructed a three–point index modeled after the one designed by H. H. Lamb (also discussed in White et al. (2018)).

**1.10   Author's Changes in Manuscript**

A three-point temperature index was created for the 1820–1842 period. This style of temperature index was first developed by H. H. Lamb (White et al., 2018). It organizes ordinal descriptions from historical records onto a three point scale. Cold conditions are given a negative value, hot conditions are given a positive value, and neutral conditions were given the value. Christian Pfister expanded on the method to create a seven–point index. However, the seven–point method requires a second dataset to validate the indexed data by. Since this is not yet available for the study region, we will be using

the three–point index here (White et al., 2018). This, combined with the rest of section 5 should accomplish this task.

Ordinal descriptions of temperature were organized into three categories: hot, neutral, and cold. Table 1 shows how the categorical temperature values were broken down into the three–point index. There were 12 values assigned to hot, 13 values were assigned to neutral, and 29 were assigned to cold. A rolling mean was taken for each day using the values for the 30 days surrounding it (the day's value, as well as the previous 15 days, and the 15 days after it). Figure 1 shows the index covering 1820–1842. The rolling mean allows the data to be smoothed without losing its daily resolution. The index is inline with with what we expected. The temperature index is at its highest during the summer months and lowest during the winter. The temperature index data is available at https://github.com/gdb12/Historical-bioclimatology.

**1.11 Comments From Referees**

-While the article does mention changing agricultural practices at Stanley Plantation (p.5, lines 15-25), the authors could do more to contextualize and emphasize the magnitude of agricultural experimentation and changes in land use in Virginia at this time, under pressure from changing markets, declining soil fertility, and new notions of agricultural "improvement." This history is discussed in more detail in another close examination of long-term Virginia plantation records (although not for purposes of historical climatology): Lynn Nelson, Pharsalia: An Environmental Biography of a Southern Plantation, 1780-1880 (University of Georgia Press, 2007). I was surprised not to see this publication in the works cited, and I believe it would be helpful for this project.

In this regard, or in section 4.2, the authors should also indicate (if possible) whether changing agricultural practices including the introduction of new cultivars appear to create artificial breaks or inhomogeneities in the timing of plant phenological observations.

[Figure]

**1.12 Author Response**

There was lot of experimentation on Shirley Plantation, including, soil treatments and new cultivars. Fortunately, these were well documented, and there were a few main cultivars that were always grown. These main cultivars can be used to create a homogeneous record, as well as allowing us to test for inhomogeneities in the expiremetal cultivars. The result is a consistent and homogeneous core of data, with experimentation being supplemental to this. The discussion was expanded to include this, and references to Nelson were added. We also added a discussion of the experimentation and changes in crops during this period to ensure that readers are aware of the possible complications.

**1.13 Author's Changes in Manuscript**

It is important note that this was a period of agricultural change and experimentation in Virginia, and Hill Carter participated heavily in these experiments (Nelson, 2007; Olmstead and Rhode, 2008b; White et al., 2018). Fortunately, he was also assiduous in documenting these efforts. He was careful to specify when he was fallowing and which fields he was doing it to. He experimented with several fertilizers and soil treatments to address soil erosion and degradation, and experimented with many cultivars. He documented when and where these experiments were conducted. He also noted when he tried new cultivars. Throughout his experiments, he used a few main cultivars every year. While the use of the various cultivars may have resulted in inhomogeneities in the plant-phenological records, there is a consistent reference variety that these can be compared to (White et al., 2018).

[...]

Tracking the cultivar is especially useful for the nineteenth century. New cultivars were developed, experimented on, and further selected for specific traits. These efforts

resulted in significant modification of the phenological and morphological traits in the crops. These modifications probably resulted in many of the crop yield gains seen in the antebellum period, as well as the expansion of the ranges of many crop species (Olmstead and Rhode, 2008a, b, 2018).

**1.14 Comments From Referees**

-The authors mention crop pests in section 4.1: does the presence of any of these also indicate specific climate conditions (similar to malaria, as discussed in the article)? If they do, it's worth mentioning; but if not, I don't recommend any changes.

**1.15 Author Response**

There are several types of pests that are sensitive to climate variation. We included a discussion of these in the new discussion in section 3. There is potential here, but it will take additional research to realize. These are now included in the discussion of the types of data available in the document we are working on.

**1.16 Author's Changes in Manuscript**

Hill Carter's plantation document records many types of observations including, but not limited to, a weather diary, plant-phenology observations, ice-phenology observations, and the phenologies of other species that are responsive to climate variation. These other phenologies include fungus, nematodes, insects, and disease causing parasites. These often came in the form of crop pests, like wheat rust, cockle, and Hessian Fly, as well as well public health factors that we now know as *Anopheles* mosquitoes and *Plasmodium vivax*.

**1.17  Comments From Referees**

-The authors may save a few lines in the discussion of challenges in early thermome-ter readings in section 4.5 by citing to Dario Camuffo, "Evidence from the Archives of Societies: Early Instrumental Observations," in The Palgrave Handbook of Climate History, ed. Sam White, Christian Pfister, and Franz Mauelshagen (London: Palgrave Macmillan UK, 2018), 83–92, $https : //doi.org/10.1057/978 - 1 - 137 - 43020 - 5_7$ and Ingeborg Auer, "Analysis and Interpretation: Homogenization of Instrumental Data," in The Palgrave Handbook of Climate History, ed. Sam White, Christian Pfister, and Franz Mauelshagen (London: Palgrave Macmillan UK, 2018), 99–105, $https : //doi.org/10.1057/978 - 1 - 137 - 43020 - 5_9$, which already summarize these issues. I offer this only as a suggestion.

**1.18  Author Response**

this section was shortened and reference given to the Palgrave Handbook. The section now focusses on the particulars of Shirley Plantation's instrumental records.

**1.19  Author's Changes in Manuscript**

There were limits to the precision of the instrument itself. Without knowing the details of its manufacture, it is not possible to tell how big of a measurement error there may or may not have been. The placement of the instrument, such as in the shade or di-rect sunlight, could also affect readings. There could be a systematic bias in how the observer was reading the instrument, such as if the mercury or alcohol in the instru-ment was between two ticks, the observer might always round it up or down. It is also possible for the observer to read the instrument differently from day to day, or for their method to change over time. The documenting of observations throughout the year

was also inconsistent. The observer might only remember to document the temperature when it is especially cold or hot. They might also forget to make the entry if they are especially busy. This can be random, or systematic. During the calving season each year, when the observer is working long and irregular hours, they might forget to enter the temperature. They might also think to make an entry in response to extreme weather events when they would not have usually remembered to. There can also be more traditional inhomogeneities in the entries. If the instrument is moved to a different location or height, a systematic bias can be introduced from suddenly being in direct sunlight or being closer to the ground. For a more detailed discussion of early instrumental records, see White et al. (2018).

1.20   Comments From Referees

-In general, I would like to see some verification of the accuracy and consistency of the climate information provided in the plantation documentation, or a suggestion how it might be verified. This verification could come from examination of its internal consistency: e.g., do variations in weather descriptions match variations in the timing of plant phenological observations? Or it could come from comparison with external information: e.g., does the Stanley Plantation document appear to consistently match other nearby records of climate or weather variability (e.g., stories about weather in a local newspaper, or descriptions of seasons in a local history) or well-observed extreme events that affected the region (e.g., the unusual winter of 1827-28 discussed in Cary J. Mock et al., "The Winter of 1827–1828 over Eastern North America: A Season of Extraordinary Climatic Anomalies, Societal Impacts, and False Spring," Climatic Change 83 (2007): 87–115, https://doi.org/10.1007/s10584-006-9126-2.)? In other words, even if the document does not provide a complete record of weather variability, is there a way to see if serves at least as a consistent high-pass filter for major variations and extremes? Even if the authors do not provide such verification here (and I do not regard it as essential for publication of this manuscript) could they suggest how it

may be done in the case of this and other plantation records? That could be helpful for future research.

**1.21   Author Response**

We feel that data validation is the topic of my next paper and is a subject large enough for its own publication. We added a section on this to our section on future work. We discuss our plans for data calibration and validation. We intend to test to see if large climate phenomena like ENSO is detectable in the data. Once an appropriate dataset is found, we will also calibrate and validate more sophisticated calibrations.

**1.22   Author's Changes in Manuscript**

Future work will focus on validating the data, expanding the database, and producing an in-depth analysis of the climate of Virginia. The most immediate concern will be data validation and calibration. We will test to see how well the data captures large scale climate phenomena like ENSO, tropical cyclone activity, and the aftermath of the Mount Tambora eruption. Calibration has proven to be more elusive. While there are a few thermometer observations in the data, they are too rare to be used for calibration. None of the local newspapers published temperature records for this period either. However, once additional plantation records from other nearby plantations, such as Berkeley Plantation (6 miles southeast of Shirley Plantation) will allow for content analysis validation (Moodie and Catchpole, 1976). Thomas Jefferson's temperature observations offer another possible dataset for validation and calibration. During his tenure as ambassador to France, Jefferson confirmed that his own thermometer was consistent with the official instrumental records in France, and maintained twice daily instrumental and ordinal weather observations during his four year residency there. this will allow us to calibrate North American records to its European contemporaries,

and validate other North American temperature records. Jefferson's own plantation records, from his garden book, have already been extracted to a database by our team (Jefferson and Baron, 1987. Work has begun on extracting Jefferson's temperature records, including his time in Paris, France (Jefferson, 1776).

**2 Referee 2**

There is much to like about this article. It explains how a source rarely used for quantitative environmental reconstructions - a plantation document - can in fact yield a wealth of remarkable data that shed light both on changes in the natural world and on social responses. The co-authors introduce a remarkable database - the product, apparently, of some 3,000 hours of work - that now has over 20,000 entries. They have the potential to make a substantial contribution to the disciplines of historical climatology and environmental history, one that certainly merits publication in a journal like Climate of the Past.

**2.1 Author Response**

Thank you.

**2.2 Comments From Referees**

Yet their article, at present, does not adequately engage with prior scholarship in historical climatology; does not explain how its database improves existing climate reconstructions; and pushes some of its conclusions beyond what its data can sustain.

First, on engagement with prior scholarship: there is, of course, a very long tradition of European and Chinese historical climatology, and the authors should clearly explain

how they are drawing from this tradition. In particular, it is striking to see no mention of Christian Pfister, whose work has made studies like this one possible. The authors attempt to describe scholarship that quantifies weather data in other documents, but their summaries a) do little justice to the sheer diversity of documents that historical climatologists have now deciphered, and b) scarcely engage with existing scholarship on the documents they mention. Any mention of ship logbook reconstructions, for example, should at the very least cite Dennis Wheeler and Dagomar Degroot.

**2.3 Author Response**

We have tried to more fully engage with the literature discussed in the comments we received. There was a substantial amount of literature that we had not been aware of. Fortunately, through the comments we received on the manuscript, we have been able to review and integrate this material. For instance, we were not familiar with Christian Pfister prior to this. In particular, The Palgrave handbook of climate history, which Pfister was an editor for, has helped us substantially improve the quality of the manuscript. We have rewritten the discussion of document types to be broader and more inclusive.

**2.4 Author's Changes in Manuscript**

Historical documents have become an important source in climate research. The breadth of document types that have been used is staggering. AS an example, researchers have used agricultural records, ship logs, port authority records, municipal records for harvest dates, newspapers, poetry, paintings, and financial reports, to just name a few. A wide array of observation types can be found in these sources. Researchers have been as creative in finding climate signals in documents as they have been in natural archives like ice cores and tree rings (White et al., 2018).

The geographical range of these sources are also diverse. Historical climatology is particularly well developed for China and Europe. By comparison, research into North America is less robust. The state of historical climatology for North America is the result of a couple of factors. Compared to Eurasian documents, written records for North America started up recently. For instance, administrative records are available for china going back two thousand years, while records for the western hemisphere mostly start up after European contact. Few records prior to the sixteenth century survived. Those records that did exist prior to 1492 were systematically purged by the Aztec Empire in c. 1380, and again by the Spanish Spanish Empire during the sixteenth century (White et al., 2018).

As European Empires colonized more of the western hemisphere, records began to be produced for these locations. What is now the Southeastern United States, excluding Florida, was one of the last areas to see permanent European colonies. These colonies were also less centralized, both in regards to population and administration (White et al., 2018). This, combined with a dispersed economy based on slavery and agriculture, means that there is comparatively few sources to work with when studying this area (Burris et al., 2018; Nelson, 2007; White et al., 2018).

**2.5   Comments From Referees**

Second, the authors should more carefully consider who is taking the observations they have databased, and why. Several additional sentences on the observers - ideally at the beginning of the article - would shed more light on how seriously we should take the observations.

**2.6   Author Response**

A description of the observer, Hill Carter, has been added, as well as a brief discussion.

**2.7 Author's Changes in Manuscript**

Hill Carter's plantation document records many types of observations including, but not limited to, a weather diary, plant-phenology observations, ice-phenology observations, and the phenologies of other species that are responsive to climate variation. These other phenologies include fungus, nematodes, insects, and disease causing parasites. These often came in the form of crop pests, like wheat rust, cockle, and Hessian Fly, as well as well public health factors that we now know as *Anopheles* mosquitoes and *Plasmodium vivax*. Carter is a particularly good source because of his naval background. While serving as a young officer during the war of 1812, he would have learned to keep consistent and accurate records. While he did not immediately apply this skill set when he took over Shirley Plantation in 1816, by 1820, he was making consistent daily entries (White et al., 2018).

The primary crops grown on the Shirley Plantation during this time as noted in the logbook included wheat, corn, and clover. Several other species were grown intermittently (see Figure 2 for a full list of crops and the years they were mentioned in the document). When and how often crops showed up in the document varied. Staple crops like wheat and corn appeared every year, while other species like beans were rarely mentioned. The set of crops remained consistent over the period but it took a few years for Hill Carter to begin regular entries concerning them, resulting in the years 1816–1819 having very few entries. The first two years of consistent entries (1820–1821) were defined by infrequent observations. For instance, missing are entries of when the first and last freezes occurred. Beginning in 1822, Carter kept more consistent documentation. Entries became more common until they were on a regular daily basis. The level of detail in the individual entries increased over this period (Figure 3).

**2.8  Comments From Referees**

More importantly, the authors should clearly explain how their database can improve - or at least supplement - existing climate reconstructions that either draw from different documents or use entirely different sources (notably paleoclimatic proxy data and model hind-casting).

**2.9  Author Response**

We have included a discussion of the contributions this database can make.

**2.10  Author's Changes in Manuscript**

This database makes contributions in a few arenas. One use of historical climate records is in conjunction with paleoclimate records, adding daily resolution data to what is available from methods like dendrochronology (Druckenbrod et al., 2003; White et al., 2018). These records can also function as a rosetta stone. Just within the Shirley Plantation's record, there are thermometer readings, extensive plant, ice, insect, and parasite phenological records, and ordinal weather observations. Once this data is calibrated and validated, it can be used to extend our understanding using these different types of data from other documents. For instance, the ice-phenological information can be combined with observations of the same type from other sources, such as other plantation records, personal diaries, travel accounts, and ship logs (The James River is navigable up to Richmond Virginia). These records also allow us to study the phenology of important agricultural species prior to the intensive modification of these species over the past two hundred years through selective breeding and, more recently, genetic modification.

This data is also unusual for when it is being produced. By the mid eighteenth century, most ordinal weather diaries had abruptly ended in favor of instrumental observations from thermometers, barometers, and other newly developed scientific instrumentation. As a result, there is very little overlap between ordinal and instrumental weather data. Most sources for historical climate data in Europe made this transition within a very short time span. This same scientific turn can be seen in many North American sources, such as in the records of Thomas Jefferson and James Madison (Druckenbrod et al., 2003; 10 Jefferson, 1776; White et al., 2018). With so little overlap to train in transfer functions, it is often difficult to calibrate and validate ordinal data (White et al., 2018). The Shirley Plantation document continues in the ordinal tradition more than a century after most other such documents end. This is concurrent with some of Hill Carter's own intermittent instrumental measurements and other such records that were produced in the area. As a result, this database represents an extraordinary opportunity to test calibration methodologies (White et al., 2018).

**2.11   Comments From Referees**

To that end, the authors really should have included a reconstruction of temperature - on an ordinal scale - developed using their database. Does that reconstruction show what we expect? What does its (presumably) high resolution do for us?

**2.12   Author Response**

We have created a three–point temperature index and added a discussion of it to section 5.

**2.13 Author's Changes in Manuscript**

A three-point temperature index was created for the 1820–1842 period. This style of temperature index was first developed by H. H. Lamb (White et al., 2018). It organizes ordinal descriptions from historical records onto a three point scale. Cold conditions are given a negative value, hot conditions are given a positive value, and neutral conditions were given the value. Christian Pfister expanded on the method to create a seven–point index. However, the seven–point method requires a second dataset to validate the indexed data by. Since this is not yet available for the study region, we will be using the three–point index here (White et al., 2018).

Ordinal descriptions of temperature were organized into three categories: hot, neutral, and cold. Table 1 shows how the categorical temperature values were broken down into the three–point index. There were 12 values assigned to hot, 13 values were assigned to neutral, and 29 were assigned to cold. A rolling mean was taken for each day using the values for the 30 days surrounding it (the day's value, as well as the previous 15 days, and the 15 days after it). Figure 1 shows the index covering 1820–1842. The rolling mean allows the data to be smoothed without losing its daily resolution. The index is inline with with what we expected. The temperature index is at its highest during the summer months and lowest during the winter. The temperature index data is available at https://github.com/gdb12/Historical-bioclimatology.

The daily resolution of this data has some advantages. Most ordinal temperature in-dices developed from historical documents are at the seasonal or annual level. Daily data will allow future research to investigate day-to-day variability. It can also be used in conjunction with daily thermometer readings and a Pfister scale to reconstruct tem-peratures at the daily, weekly, or monthly temperatures. This can then be aggregated to seasonal and annual levels so it can be used in conjunction with other reconstructions (White et al., 2018).

**2.14  Comments From Referees**

Third, the section on malaria is poorly introduced. Terms and key relationships are referenced offhand before they are explained, and that will need some reworking. More importantly, it seems to me that the authors too directly link temperature to the first cases of malaria on the plantations. Of course, temperature has a major impact on malaria, but it is far from the only variable to consider. Plantations were artificial hot spots for malaria, and annual changes in first cases could well reflect human actions, decisions, and programs. Malaria is also very sensitive to rainfall, particularly alternating droughts and deluges (as John McNeill, among many others, has explained). At the very least, such complexity should be thoroughly acknowledged and articulated.

**2.15  Author Response**

We have altered the discussion of malaria in a few ways. We reorganized descriptions and definitions so that they appear when terms are first introduced. We have qualified our conclusion by adding a discussion of other environmental factors that could be affecting the timing of infections.

**2.16  Author's Changes in Manuscript**

It is important to recognize that temperature is not the only limiting factor in the development of *Anopheles* and *Plasmodium*. For instance dry conditions can leave more water stagnating, providing ideal development conditions for the larva and pupa stages of mosquitoes. Large amounts of precipitation on the other hand can wash away water, making it difficult for mosquitoes to complete their life cycles (Guerra et al., 2008; Gething et al., 2011; White et al., 2018).

Human activity can also alter the life cycles of *Plasmodium* and *Anopheles*. Even factors like proximity of dwellings and work spaces to areas where *Anopheles* eggs and larva develop can impact infection rates and timings. The timing of labor activities can also influence this. For instance, people on Shirley would be much more likely to get infected while working on reclaimed swamp land. This part of the plantation had trouble dealing with standing water and was in close proximity with the remaining swamp land (Stampp et al., 1985). Without having controlled for these other environmental conditions, this analysis can only be suggestive of changes in climate (Gething et al., 2011; Guerra et al., 2008; McNeill, 2010).

**2.17 Comments From Referees**

Finally, the article is, at present, rife with baffling typos and grammatical flaws - ones that should have been weeded out well before submission. It also contains some unintentionally comedic passages, such as: "summer temperatures were entered in the nineties, making it most likely that measurements were in degrees Fahrenheit rather than degrees Celcius" (and again, note the typo). Obviously these problems should be corrected for the revision. In short: the article should be published, in my opinion, but only after major revisions

**2.18 Author Response**

We have cleaned up the grammar and tightened up the prose as well as giving it a more thorough proofing.

**2.19 Author's Changes in Manuscript**

[Changes were scattered throughout the manuscript]

While the units were not explicitly stated, summer temperatures were recorded as being as high as the nineties, clearly indicating that measurements were in degrees Fahrenheit rather than degrees Celsius.

[Figure]

**3  New Figure and table**

**Fig. 1.** Three-point temperature index for the period 1816–1842. See text for discussion of how the index was constructed.

indexdaily.pdf

**Table 1.** Temperature index Categories.

| Hot | Neutral | Cold |
|---|---|---|
| exceptionaly hot | fair | below freezing |
| excessively hot | fine | cold |
| hot | mild | cool |
| hotest day this year | mild and pleasant | exceedingly cold |
| hotest morning this year | milder | excessively cold |
| indian summer | mildest winter ever known this far | excessively cold for season |
| sultry | moderate | first frost |
| very hot | nice | freeze |
| very warm | pleasant | frost |
| very warm for season | very mild | frost very slight |
| violently hot | warm | frozen |
| workers fainted from heat | warm and pleasant | ground frozen |
| | warmer | ice |
| | | light frost |
| | | river frozen |
| | | river hard frozen |
| | | river nearly frozen |
| | | second frost |
| | | severe frost |
| | | severe white frost |
| | | severest frost yet |
| | | smart frost |
| | | turned colder |
| | | turning cold |
| | | very cold |
| | | very cold violent change |
| | | very cool |
| | | white frost |
| | | wintery |